# Recurrent breakpoints in the *BRD4* locus reduce toxicity associated with gene amplification

## Graphical abstract

## Authors

Jeremiah Wala, Simona Dalin, Sophie Webster, ..., Rameen Beroukhim, Pratiti Bandopadhayay, Veronica Rendo

## Correspondence

veronica.rendo@igp.uu.se

## In brief

Wala et al. identify recurrent focal *BRD4* deletions in tumors harboring larger amplifications, suggesting that these deletions may serve to limit toxic *BRD4* overexpression. A CRISPR-Cas9 cell line model validates that fine-tuned *BRD4* expression is required for cancer cell growth. Together, these support the first recurrent deletion reducing toxicity in cancer.

## Highlights

- Focal deletions of *BRD4* regulatory regions are found in multiple cancers

- *BRD4* deletions normalize *BRD4* expression in tumors with larger chromosome 19 gains

- In a cell line model, focal *BRD4* deletions rescue gene overexpression toxicity

- Fine-tuned levels of *BRD4* expression are required for sustained tumor proliferation

 Wala et al., 2025, Cell Genomics *5*, 100815
April 9, 2025 © 2025 The Author(s). Published by Elsevier Inc.

# Cell Genomics

CellPress

## Article

# Recurrent breakpoints in the *BRD4* locus reduce toxicity associated with gene amplification

Jeremiah Wala,[1,2,5] Simona Dalin,[1,2,5] Sophie Webster,[1,2,5] Ofer Shapira,[1,2] John Busanovich,[1] Shahab Sarmashghi,[2] Rameen Beroukhim,[1,2] Pratiti Bandopadhayay,[2,3,6] and Veronica Rendo[1,2,4,6,7,*]

[1]Departments of Cancer Biology and Medical Oncology, Dana-Farber Cancer Institute, Boston, MA 02215, USA
[2]Broad Institute of MIT and Harvard, Cambridge, MA 02142, USA
[3]Department of Pediatric Oncology, Dana-Farber Cancer Institute, Boston, MA 02215, USA
[4]Department of Immunology, Genetics, and Pathology, Uppsala University, 75185 Uppsala, Sweden
[5]These authors contributed equally
[6]These authors contributed equally
[7]Lead contact
*Correspondence: veronica.rendo@igp.uu.se

## SUMMARY

Recent work by the ICGC-PCAWG consortium identified recurrent focal deletions in the *BRD4* gene, decreasing expression despite increased copy number. We show that these focal deletions occur in the context of cyclin E1 amplification in breast, ovarian, and endometrial cancers, and serve to disrupt *BRD4* regulatory regions and gene expression across isoforms. We analyze open reading frame screen data and find that overexpression of BRD4 long (BRD4-L) and short isoform BRD4-S(a) impairs cell growth across cell lines. We confirm these results in OVSAHO ovarian cancer cells, where the overexpression of *BRD4* isoforms significantly reduces tumor growth. Next, we mimic *BRD4* focal deletions using CRISPR-Cas9 technology and show that these focal deletions rescue ovarian cancer cells from toxicity associated with BRD4 overexpression, suggesting that BRD4 levels must be fine-tuned for cancer cell proliferation. Our study provides experimental evidence for the first recurrent deletion reducing toxicity in cancer, expanding the landscape of cancer progression mechanisms.

## INTRODUCTION

Large-scale whole-genome sequencing efforts have increasingly highlighted the role of genomic rearrangements in driving cancer progression. These structural variants (SVs) involve the ligation of genomically distant DNA regions and can result in deletions, inversions, duplications, or more complex events. SVs can regulate gene expression by altering copy-number status, affecting nearby regulatory elements, or enabling novel fusion products.[1–5] Altogether, SVs represent a common mechanism by which cancer cells activate oncogenes (e.g., formation of *IGH-BCL2* and *KIAA1549-BRAF* fusions) or disrupt the function of tumor suppressor genes (e.g., *TP53*, *PTEN*).

A recent study from the International Cancer Genome Consortium's (ICGC) Pan-Cancer Analysis of Whole Genomes (PCAWG) described the landscape of SVs commonly occurring in 30 tumor types by analysis of 2,658 genomes.[3] From this effort, focal SVs within the bromodomain-containing protein 4 (BRD4) in chromosome 19p were detected in a fraction of ovarian (n = 8) and breast (n = 7) cancers, occurring primarily in tumors with amplifications spanning the *BRD4* locus. These breakpoints resulted in small, focal deletions overlapping with intron 1 and often exon 1 of *BRD4*. Interestingly, tumors exhibiting these SVs had constant or decreased levels of *BRD4* gene expression relative to their

copy number. This was a striking observation, as increases in copy number for most genes tend to translate into increased gene expression.[6] This disruption in gene dosage was hypothesized to constitute an undescribed mechanism to fine-tune gene overexpression in tumors subject to *BRD4* genomic amplification. These *BRD4* focal deletions appear to be the first example of a driver SV alteration that reduces toxicity in cancer, but the functional impact of these focal deletions remains unknown.

BRD4 is a member of the bromodomain and extra-terminal domain family of proteins, able to bind acetylated histones to regulate gene transcription. Key biological roles include maintenance of chromatin structure to ensure epigenetic memory after mitosis and transcriptional regulation of signal-inducible genes by association with the positive transcription elongation factor (p-TEFb) complex and RNA polymerase II.[7–9] *BRD4* is commonly altered in human cancers, and significant amplification is observed together with the known oncogene cyclin E1 (*CCNE1*) in both ovarian and breast carcinomas.[10,11]

In this study we sought to determine the impact of *BRD4* recurrent focal deletions on gene expression, with a particular focus on how *BRD4* isoform-level expression is altered in *BRD4*-amplified ovarian, breast, and endometrial tumors. We additionally leverage functional genomic tools to assess the cellular effect of dysregulated *BRD4* expression across multiple

**Cell Press**

human cancer cell lines. This validates the identified focal deletions in an ovarian cell line model as a mechanism to sustain cellular proliferation while dampening the toxicity effects associated with initial gene amplification. Our findings provide the first experimental evidence for an undescribed mechanism by which genetic alterations reduce copy-number-associated toxicity to drive cancer.

## RESULTS

### Significantly recurrent breakpoints result in focal deletion of *BRD4* regulatory regions

To better understand somatic breakpoint distribution at the *BRD4* locus, we first accessed whole-genome sequencing data from the PCAWG cases where recurrent rearrangements were originally identified.[3] We looked at breakpoint densities present in all PCAWG cases and confirmed that there is an enrichment of breakpoints within a small region of *BRD4*, typically ranging from exon 1 to intron 1 of the gene (Figure 1A). We were further able to identify similar patterns of rearrangements when comparing these samples to an additional breast cancer dataset[12] (*n* = 560 whole-genome sequences), supporting the notion that this may constitute a mutational event with functional implications in cancer.

To understand the effect of these recurrent breakpoints, we next analyzed the PCAWG somatic copy-number alteration (SCNA) data to look for SCNA events at the *BRD4* locus corresponding to the recurrent breakpoints. This revealed recurrent focal deletions (<100 kbp) in *BRD4* in 16 tumors, including 4 breast cancers, 7 ovarian, 3 endometrial and 2 colorectal cancers (Figure S1). We also identified an additional 12 cases involving a copy loss of *BRD4* relative to the neighboring *NOTCH3* gene (1 breast, 3 ovarian, 2 endometrial, 3 pancreatic adenocarcinoma, 1 pancreatic endocrine, 1 lung squamous, and 1 hepatocellular cancers; Figure S1), although these were not strictly focal (<100 kbp) deletions. The total copy number in genes adjacent to the *BRD4* locus tended to be higher in samples with focal *BRD4* deletions than in samples without *BRD4* deletions (*NOTCH3* median total copy number: 3.3 vs. 2.0; $p < 8 \times 10^{-9}$, Wilcoxon rank-sum test). Finally, we identified an additional 6 samples (3 ovarian, 2 breast, and 1 endometrial) with breakpoints within the recurrent region of *BRD4*, but with no corresponding SCNA. On further inspection of the raw read-depth signal, these samples each exhibited a read-depth drop within the boundaries of the breakpoints and with breakpoint orientations consistent with a focal deletion (Figure S2). Thus, we conclude that these also represent focal *BRD4* deletions given the commensurate read-depth and breakpoint signal, despite not having a sufficiently strong signal with read-depth alone to trigger an SCNA call. We therefore identified a total of 22 tumors (6 breast, 10 ovarian, 4 endometrial, and 2 colorectal) with *BRD4* focal deletion for further analysis.

The enrichment for focal *BRD4* deletions in breast, ovarian, and endometrial cancers was striking (*p* < 0.02 for all three tumor types, *p* < 0.005 for each alone; Fisher's exact test). We hypothesized that the increased *BRD4* copy-number in these samples may be due to large-scale amplifications involving cyclin E1 (*CCNE1*), a known driver in endometrial, breast, and ovarian can-

cers, which is also located on chromosome 19. Considering all tumor types, the mean copy number at *CCNE1* was significantly higher in *BRD4*-deletion samples compared to samples without *BRD4* deletions (6.7 vs. 3.1; $p = 2.6 \times 10^{-8}$; Wilcoxon rank-sum test; Figure 1B). We similarly observed an increase in the mean *CCNE1* copy number when considering only breast, ovarian, and endometrial cancers with and without a *BRD4* focal deletion (7.1 vs. 4.1; *p* = 0.0002; Wilcoxon rank-sum test; Figure 1C). Additionally, tumors with amplicons containing the *BRD4* locus (*BRD4*-amplified) had significantly more concomitant focal deletions in *BRD4* compared to those without gene amplification (*p* = 0.00004) (Figure 1D). In each tumor type, areas of genomic deletion overlapped with epigenetic marks of transcriptionally active chromatin, including H3K4me3 peaks from ovarian (HMEC) and breast adenocarcinoma (MCF-7) cell lines[14] (Figure 1A). We therefore hypothesized that focal *BRD4* deletions would lead to the decreased expression of *BRD4*, either through copy-number dosage effects or to disruption of key regulatory regions, in tumors with larger-scale amplifications involving both *BRD4* and *CCNE1*.

We further hypothesized that selective pressures may favor a particular temporal ordering to the chromosome 19 large-scale amplification and *BRD4* focal deletion events. To address this, we analyzed allelic copy-number data from PCAWG tumors to determine the most parsimonious order of events. Loss of heterozygosity on an otherwise amplified allele suggests a primary deletion, whereas an amplified allele with copy loss to a nonzero number implies the amplification likely occurred first (see STAR Methods). Six cases were ambiguous due to insufficient allelic information in the case of very short deletions or in the case of deletions affecting a different allele than the amplification. In the 13 samples with focal deletions in *BRD4* and sufficient allelic information, we identified an enrichment for amplifications occurring first (in 11 samples) compared with deletions occurring first (2 samples; *p* = 0.01; binomial test; Figure S3). This suggests that *BRD4* focal deletions may confer an additional fitness advantage to *CCNE1*-amplified tumors, rather than serve as a necessary antecedent to these amplifications.

### Focal deletions of *BRD4* significantly decrease *BRD4* expression across isoforms

We next performed differential expression analysis to identify the genes most dysregulated in tumors with *BRD4* focal deletions. Controlling for tumor type and restricting to autosomal protein-coding genes, we identified 22 genes (of 18,061 genes) with significantly differential expression between the *BRD4* focal deletion and non-deleted cohorts (*q* < 0.1 cutoff with Bonferroni correction and at least 2-fold expression change; Figure 2A). Of these 22 genes, 17 were overexpressed in the *BRD4* focal deletion group. Interestingly, 76% (13/17) of overexpressed genes were located on chromosome 19, including *CCNE1*. *BRD4* did not achieve significance, even in a larger list of 584 genes that achieved significance when performing less-restrictive false discovery rate correction. We also performed differential expression analysis without controlling for tumor type, finding *CCNE1* to be the single most significant gene in this analysis and with 57% (17/30) of the top 30 genes being found on chromosome 19. Again, there was no significant difference in *BRD4*

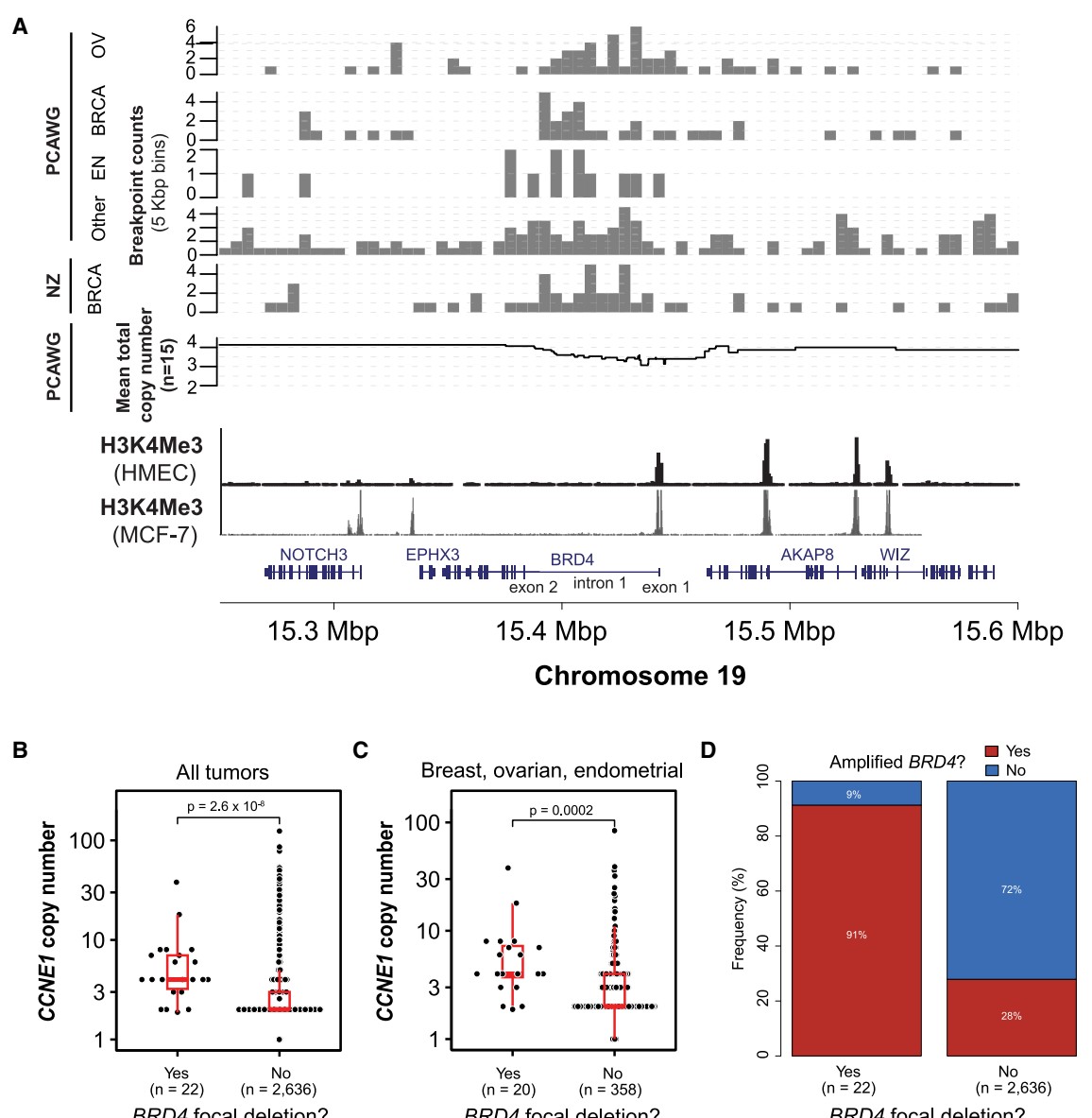

**Figure 1. Significantly recurrent breakpoints result in focal deletion of *BRD4* regulatory regions**

(A) Top: breakpoint densities for PCAWG breast (BRCA), endometrial (EN) and ovarian (OV) cancers exhibiting rearrangements at the *BRD4* locus, compared to an independent breast cancer dataset (Nik-Zainal et al.[12]); Center: somatic copy-number alterations (SCNAs) for pooled breast, ovarian, and endometrial cancer samples with *BRD4* focal deletions. Bottom: alignment with H3K4me3 from breast (MCF-7) and ovarian (HMEC) cell lines. Genomic track: gene exons and intron as defined in RefSeq[13] for hg19.

(B and C) (B) Total copy number at *CCNE1* locus (chromosome 19) between tumors with or without *BRD4* focal deletions and (C) only breast, ovarian, and endometrial tumors; Wilcoxon rank-sum test. The box spans the interquartile range (IQR) with the median as a horizontal line and bars extending to 1.5 × IQR.

(D) Proportion of tumors with or without amplifications spanning the *BRD4* locus, comparing those with or without a concomitant *BRD4* focal deletion.

expression. We thus reasoned that *BRD4* focal deletions prevent toxic *BRD4* overexpression that would have otherwise occurred in the background of chromosome 19 amplifications driving *CCNE1*.

*BRD4* encodes two splice variants, "long" and "short," with distinct cellular localization and biological roles. The long isoform (BRD4-L; Ensembl: BRD4-201) consists of 20 exons and en-

codes a 1,362-amino acid protein mostly confined to the nuclear membrane. It has both a proline-rich region and a C-terminal motif that allows for interaction with the p-TEFb complex and RNA polymerase II.[15] In contrast, the BRD4 "short" isoform (BRD4-S) encodes for a dominant (BRD4-S(a); Ensembl: BRD4-203) and a secondary (BRD4-S(b); Ensembl: BRD4-202) protein. BRD4-S(a) consists of the first 12 exons, encoding a

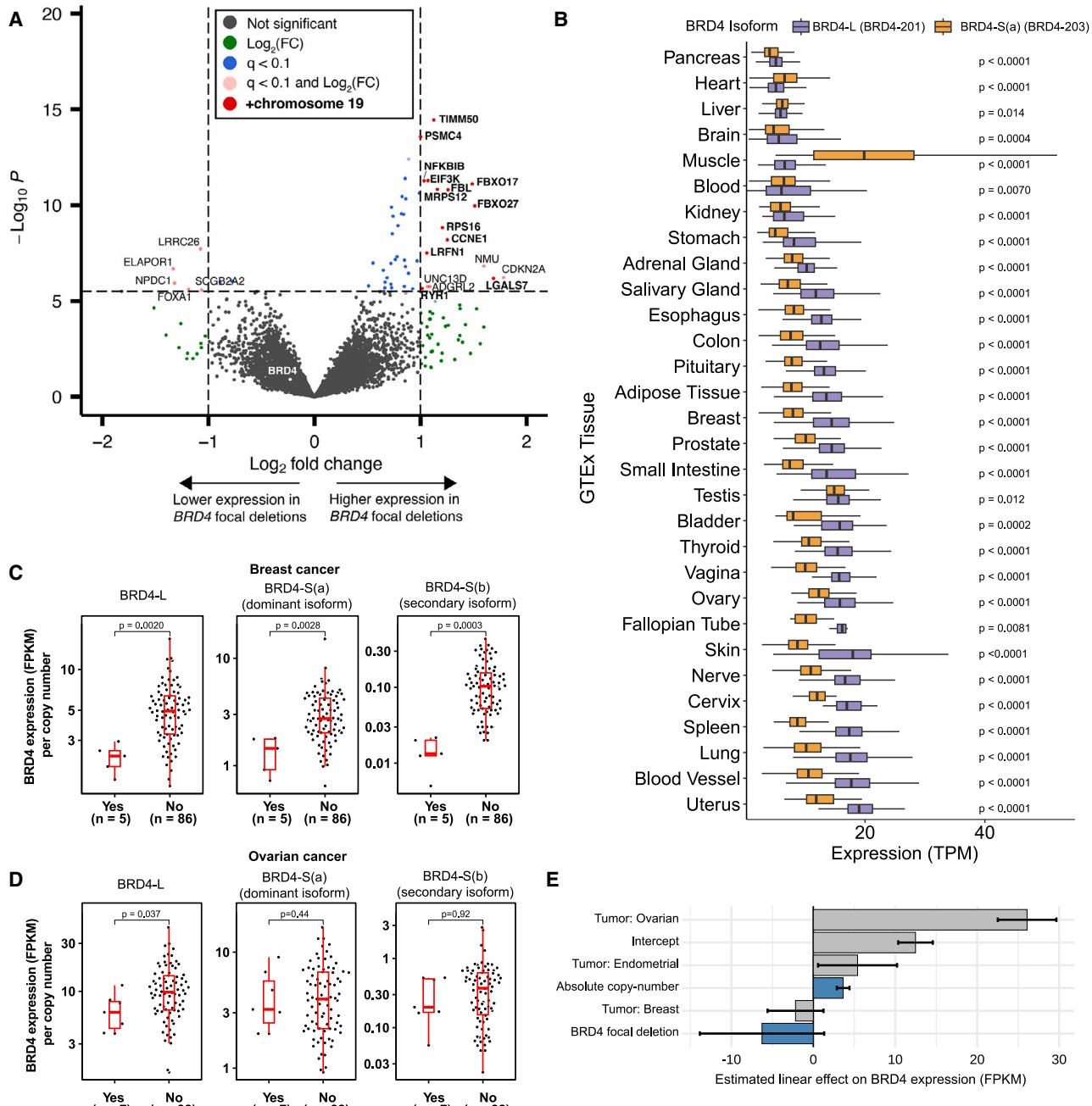

**Figure 2. Focal deletions of *BRD4* significantly decrease *BRD4* expression across isoforms**

(A) Volcano plot of global gene expression comparing *BRD4* focal deletion tumors (*n* = 22) with all other PCAWG tumors (*n* = 2,636), Bonferroni-corrected cutoff of *p* < 0.1. Genes on chromosome 19 are displayed in bold type.

(B) Expression of BRD4-L (ENST00000263377; BRD4-201) and BRD4-S(a) (ENST00000371835; BRD-203) isoforms in normal tissues as obtained from GTEx (TPM, transcripts per million). Two-tailed Wilcoxon signed rank test with Bonferroni-corrected *p* values as indicated.

(C and D) Isoform-specific expression of *BRD4* in PCAWG breast cancer (C) and ovarian (D) tumors, normalized to copy number, comparing BRD4-L (ENST00000263377; BRD4-201) with two BRD4-S isoforms (dominant BRD4-S(a): ENST00000371835, BRD-203, and secondary BRD4-S(b): ENST00000360016, BRD4-202) between tumors with or without *BRD4* focal deletions (FPKM, fragments per kilobase of transcript per million fragments mapped). Wilcoxon rank-sum test. The box spans the IQR with the median as a horontal line and error bars extending to 1.5.

(E) Coefficients of a linear model for *BRD4* expression controlling for tumor type (non-breast, ovarian, or endometrial as baseline), absolute copy number, and focal deletion status (not deleted as baseline). Error bars represent the 95% confidence intervals for coefficient estimates from the linear model.

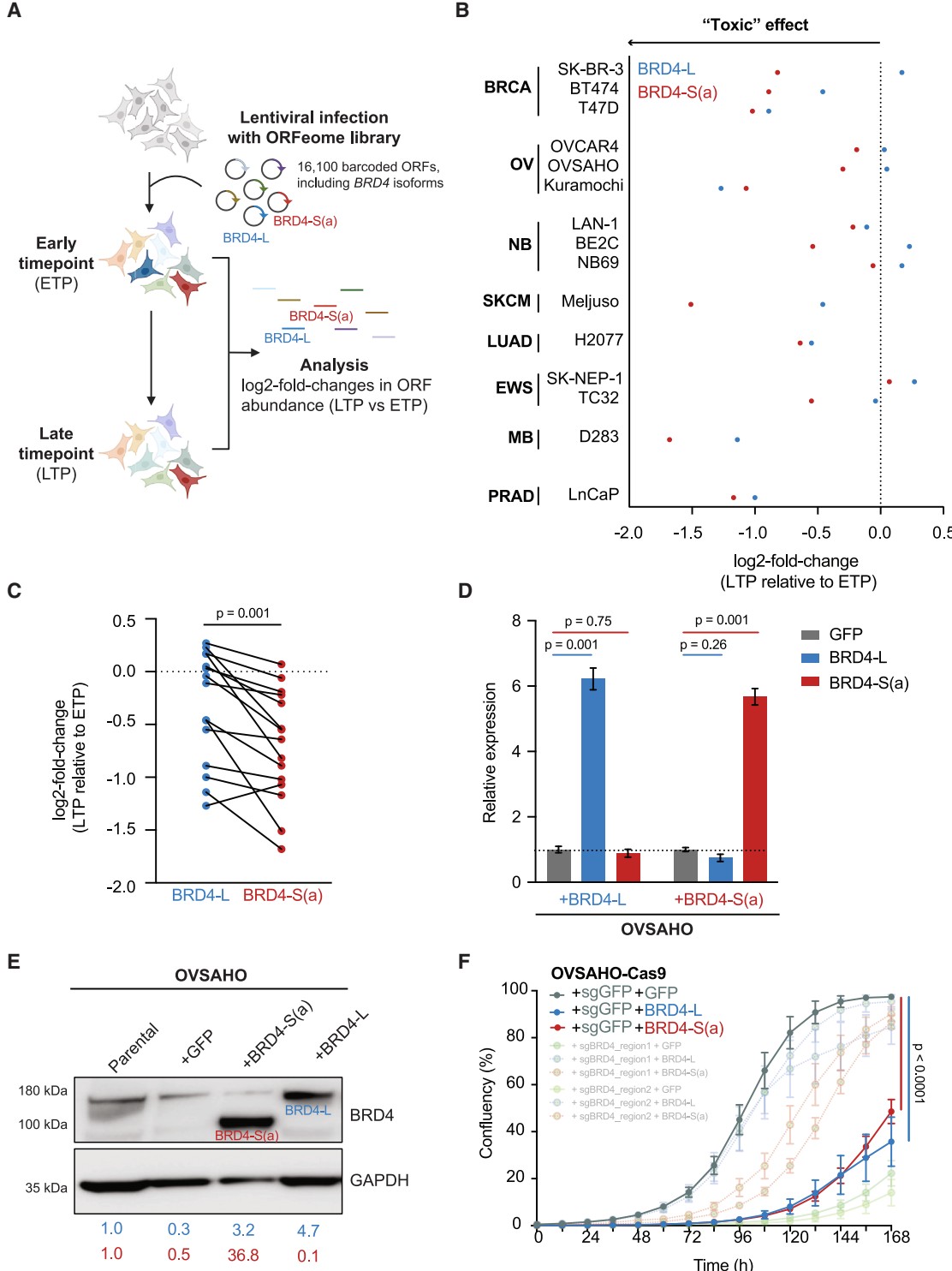

**Figure 3. *BRD4* overexpression is toxic across multiple cancer types**

(A) Overview of an ORF screen, from which the effect of BRD4-L and BRD4-S(a) overexpression on cell proliferation can be assessed.

(B) Log$_2$-fold changes in cell proliferation for BRD4 constructs, expressed relative to the experimental early time point (ETP) of each screen. Negative changes reflect a detrimental effect on cell proliferation. ORF screens were conducted in breast (BRCA), ovarian (OV), neuroblastoma (NB), skin cutaneous melanoma (SKCM), lung adenocarcinoma (LUAD), Ewing sarcoma (EWS), medulloblastoma (MB), and prostate adenocarcinoma (PRAD) cell lines.

(C) ORF screen performance between BRD4-L and BRD4-S(a) isoforms for each cell line. Two-tailed t test.

*(legend continued on next page)*

722-amino acid protein located in the nuclear matrix. While lacking the proline-rich domain, this isoform modulates chromatin organization and recruitment of transcription factors.[16] The secondary BRD4-S(b) isoform, with an alternative first exon, encodes a 794-amino acid protein. In the PCAWG tumors, the secondary short BRD4-S(b) isoform was significantly less expressed compared to the dominant BRD4-S(a) short isoform (median fragments per kilobase of transcript per million fragments mapped; FPKM - BRD4-S(a): 5.89; BRD4-S(b): 0.20; $p < 10^{-16}$, Wilcoxon rank-sum test). In cancer, BRD4-L has been attributed a tumor-suppressive role, while BRD4-S is deemed oncogenic and reported to be highly expressed in metastatic disease.[17,18] We therefore asked whether the deletions' effects on expression favored one of these two isoforms.

We first assessed the distribution of BRD4-L and primary BRD4-S(a) isoforms across normal tissues, using the Adult Genotype Tissue Expression (GTEx) Project (Figure 2B). In nearly every tissue type, BRD4-L is expressed at significantly higher levels ($p < 0.0001$; Wilcoxon signed-rank test) than BRD4-S(a), except for heart, liver, muscle and blood tissues. Given the direct implications for oncogenesis, we conducted analyses in the PCAWG samples to determine whether focal deletions occurring on the regulatory regions of *BRD4* could impact, in addition to global gene expression, the ratio of expressed isoforms. When normalized for locus-specific copy number, we identified significantly decreased copy-number-adjusted *BRD4* expression of BRD4-L and both isoforms of BRD4-S in breast cancer tumors with *BRD4* focal deletions (Figure 2C). In ovarian cancer, each of the three isoforms was decreased in the *BRD4* focal deletion tumors, although only the BRD4-L isoform achieved statistical significance ($p = 0.037$; Wilcoxon rank-sum test; Figure 2D). Taken together, these data provide evidence of how *BRD4* focal deletions alter gene expression across isoforms and ultimately reduce the overexpression of a gene residing in a genomically amplified region.

We next asked to what extent *BRD4* focal deletions alter *BRD4* expression relative to the copy number of the larger amplicon and to the tumor type of origin. We used linear regression to model the effects of copy number, *BRD4* focal deletion status, and tumor type on *BRD4* expression. Tumor type had the largest effect on *BRD4* expression (measured in FPKM), with ovarian cancer having the largest positive effect (regression coefficient: 26.1, 95% confidence interval [CI]: 22.5–29.7; Figure 2E). As expected, copy number further increased *BRD4* expression (regression coefficient: 3.6, 95% CI: 2.9–4.4). However, focal deletions decreased *BRD4* expression by nearly as much as two units of copy loss (regression coefficient: −6.3, 95% CI: −13.8 to 1.3). These data identify that the absolute decrease in expression due to *BRD4* focal deletions remains

significant even when controlling for tumor type and copy number, and that the extent to which absolute *BRD4* expression levels are inherently oncogenic or toxic is highly dependent on tissue of origin.

### *BRD4* gene overexpression is detrimental for cancer cell proliferation

We hypothesized that *BRD4* is initially amplified in cancer cells, but high expression levels are not tolerated. *BRD4* gene overexpression has been linked with a growth-inhibition phenotype in HeLa cervical cancer cells,[19] but the effects of gene overexpression across multiple cancer types has not been systematically interrogated. We therefore compiled data from the control arm of 15 different open reading frame (ORF) overexpression screens performed in 8 tumor types,[20–28] where the effects of gene overexpression on cell growth can be quantified over time. We next determined the log-fold changes in cell growth following overexpression of BRD4-L and BRD4-S(a) isoforms across different cell lines for a period of 2–3 weeks and in the absence of any additional selection pressures (Figure 3A). In 10/16 cell lines (62.5%), global *BRD4* overexpression decreased cell growth (as evidenced by negative log-fold changes), supporting its role as a "toxic" gene in cancer.[29] For the remaining cell lines, gene overexpression had a neutral effect on cell proliferation (Figure 3B). Within each cell line, overexpression of BRD4-S(a) was significantly more toxic ($p = 0.001$) than BRD4-L (Figure 3C).

We further selected OVSAHO ovarian carcinoma cells as a model to validate the toxic effects associated with *BRD4* overexpression from the ORF screen data. We focused on ovarian tumors as they present higher *BRD4* gene amplification rates (~12%) when compared to breast cancers (~3%).[30] First, we stably transduced OVSAHO cells with a lentiviral vector encoding BRD4-L, BRD4-S(a), or GFP control. Following cell selection, we validated gene overexpression by quantitative reverse-transcription PCR (RT-qPCR) using isoform-specific primers. We were able to detect >5-fold increases in BRD4-L and BRD4-S(a) mRNA transcripts ($p < 0.001$) when compared to GFP-transduced OVSAHO cells (Figure 3D). These changes translated into BRD4 protein expression levels, where BRD4-L and BRD4-S(a) isoforms were expressed 4.6- and 36.8-fold higher, respectively, compared to GFP controls (Figure 3E). We confirmed BRD4-L protein expression with an isoform-specific antibody (Figure S4). To assess the effects of *BRD4* overexpression on cell proliferation, we conducted live-cell image analysis to quantify increases in confluency present in cells overexpressing different isoforms over the course of 7 days (Figures 3F and S5). Overexpression of both BRD4-L and BRD4-S(a) mRNA isoforms led to a profound growth-inhibitory phenotype ($p < 0.001$) compared to GFP overexpressing cells, confirming that supraphysiological

(D) Validation of BRD4 isoform expression in OVSAHO cells. Transcript expression is quantified relative to GFP control. Mean and standard deviation of three replicates; two-way ANOVA.

(E) Immunoblot of BRD4-L and BRD4-S protein variants detected in OVSAHO cells transduced with BRD4 or control GFP overexpression constructs. Values represent levels of protein expression, normalized to glyceraldehyde 3-phosphate dehydrogenase expression and shown relative to parental cells (endogenous BRD4 levels).

(F) Quantification of cell confluency in OVSAHO-Cas9 after transduction with BRD4-L, BRD4-S(a), or GFP control overexpressing constructs. Mean and standard deviation of three replicates, two-way ANOVA followed by Dunnett post-tests, controlling the Family-wise alpha threshold and confidence level. *p* values of the final time point are annotated.

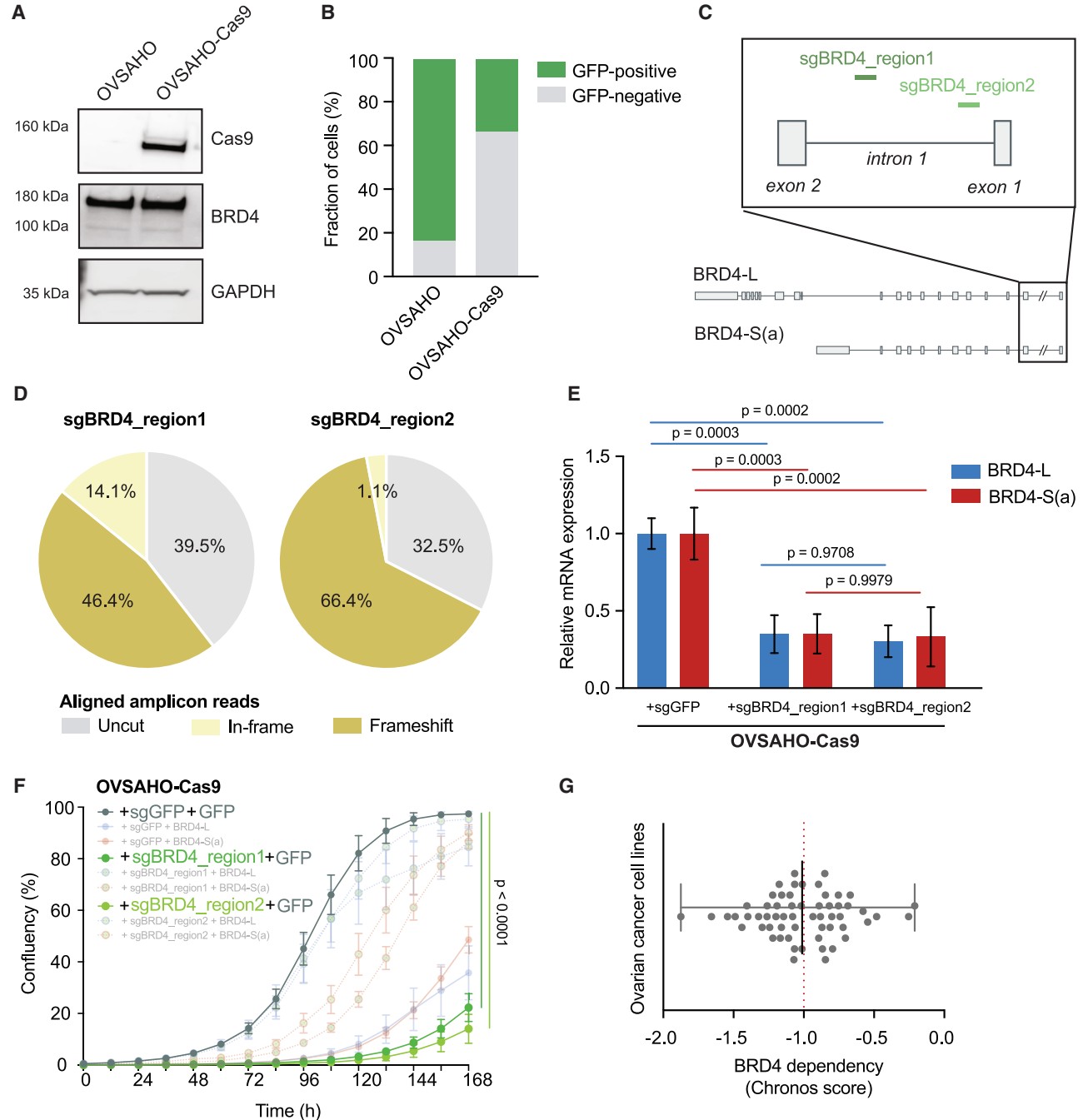

**Figure 4. Use of CRISPR-Cas9 technology to mimic the effect of *BRD4* focal deletions**

(A) Confirmation of Cas9 expression in OVSAHO cells by immunoblotting.

(B) Fraction of GFP[+] cells detected in Cas9 activity assay. To measure Cas9 cutting efficiency, OVSAHO-Cas9 cells were transduced with a vector encoding GFP and sgGFP. After 10 days, a decreased population of GFP[+] cells was detected by flow cytometry, confirming Cas9 activity.

(C) Intronic *BRD4* regions targeted by CRISPR-Cas9 sgRNAs (inset not to scale due to length of intron 1).

(D) CRISPR sequencing results for sgBRD4_region1 and sgBRD4_region2, showing the percentage of uncut, frameshift, and in-frame *BRD4* reads when aligned to a reference sequence.

(E) Expression of *BRD4* isoforms following CRISPR-Cas9-mediated sgRNA cutting, relative to sgGFP control. Median and standard deviations of three replicates, two-way ANOVA.

*(legend continued on next page)*

*BRD4* gene expression levels have a negative impact on cellular fitness.

### *BRD4* focal deletions rescue ovarian carcinoma cells from toxicity effects associated with gene overexpression

The presence of focal deletions spanning *BRD4* regulatory regions suggests that these rearrangements may mechanistically reduce gene expression to levels that are no longer toxic to cancer cells. To functionally validate this hypothesis, we first stably infected OVSAHO cells with a lentiviral Cas9 vector and confirmed protein expression (Figure 4A) and activity (Figure 4B) 10 days following selection. Second, we transduced cells with CRISPR-Cas9 single-guide RNAs (sgRNAs) designed to target two intronic regions (intron 1 of RefSeq[13]: NM_001379291) of *BRD4* ("sgBRD4_region1" and "sgBRD4_region2"), mimicking the effect of the focal deletions originally identified in PCAWG samples (Figure 4C). We were able to confirm sgRNA cutting by CRISPR sequencing (Figure 4D), detecting 46.4% of frameshift cuts for sgBRD4_region1 and 66.4% for sgBRD4_region2 when amplicon reads were aligned to a reference *BRD4* sequence. To assess global and isoform-specific levels of *BRD4* expression following sgRNA cutting, we performed an RT-qPCR in OVSAHO-Cas9 cells transduced with *BRD4* sgRNAs relative to an sgGFP control (Figure 4E). Global *BRD4* mRNA levels were decreased by ∼40% following sgRNA cutting ($p < 0.0001$). No difference in the ratio of BRD4-L to BRD4-S(a) isoforms was detected when regions 1 or 2 of *BRD4* were deleted ($p > 0.99$). Given the overlap in peptide sequence between BRD4 protein isoforms, we did not assess the effects of sgRNA cutting on protein expression. Next, we evaluated whether deletion of these intronic *BRD4* regions impacts cell proliferation. By live-cell imaging, we were able to confirm that deletions by sgBRD4_region1 and sgBRD4_region2 significantly decrease cell confluency by ∼50% at 7 days compared to cells transduced with an sgGFP control ($p < 0.0001$) (Figures 4F and S5). The observation that *BRD4* ablation is detrimental to OVSAHO cell growth is in agreement with CRISPR-Cas9 dependency data from the Dependency Map Consortium (DepMap Public 23Q4 release, http://depmap.org), where *BRD4* scores as a genetic dependency (mean Chronos score = −0.99 ± 0.28) and is essential in 33/59 (∼56%) of ovarian cancer cell lines (Figure 4G).

We finally attempted to functionally recreate the gene regulation patterns observed in *BRD4*-amplified PCAWG tumors that undergo focal rearrangements at the *BRD4* locus. This would allow us to confirm that the observed focal deletions indeed reduce and fine-tune levels of *BRD4* expression to rescue cells from gene toxicity and ultimately sustain proliferation. To time *BRD4* overexpression with *BRD4* deletion, we first transduced OVSAHO Cas9-expressing cells with sgBRD4_region1 and sgBRD4_region2, and later introduced BRD4-L or BRD4-S(a) according to previously established experimental time points (Figure 5A). After 7 days, we observed that the deletion of sgBRD4_region1 and sgBRD4_region2 rescued proliferation of BRD4-L and BRD4-S(a) OVSAHO cells compared to an sgGFP control (Figures 5B and S5). Our results point to a "Goldilocks" model of *BRD4* gene regulation, where too-high or too-low levels of expression have a negative effect on cellular fitness. In *BRD4*-amplified tumors, focal deletions in gene regulatory regions serve as a mechanism to decrease and restore gene expression to levels that are tolerated and beneficial for proliferation (Figure 5C). This represents an undescribed mechanism of gene-dosage compensation in human cancers.

### DISCUSSION

The distribution of rearrangements across the cancer genome not only reflects the mechanisms that give rise to their formation (e.g., homologous recombination, microhomology-dependent recombination, retrotransposition) but also the advantage they confer for cellular fitness.[31,32] PCAWG's initial analysis of non-coding somatic drivers in over 2,600 cancer whole genomes identified breast and ovarian tumors with recurrent breakpoints in the *BRD4* locus on chromosome 19p, which result in focal deletions that span gene regulatory regions and reduce global expression levels.[3] Our present study reveals that these rearrangements occur in the context of *CCNE1* amplification and serve to decrease *BRD4* expression across isoforms to limit its copy-number-driven overexpression. Our finding that these focal deletions occur in tumors with larger chromosome 19 amplifications further supports the hypothesis that *BRD4* focal deletions serve as a mechanism to fine-tune gene expression, rather than alter a tumor suppressor gene, and is more often observed with recurrent deletions. We have also provided functional evidence for this effect in an ovarian cancer cell line model.

*BRD4* overexpression has been previously linked with the suppression of cancer cell growth in selected human and mouse cell line models,[19] but it has not been attributed to the toxicity profile we observed when systematically interrogating the effects of gene overexpression across multiple tumor types. These toxicity patterns seem to additionally differ between BRD4-L and BRD4-S isoforms. This suggests the importance of characterizing which *BRD4* variants (both described here and beyond our study) become deregulated in a particular cancer type, as they can exert oncogenic or tumor suppressive roles (in an isoform-specific manner) and may inform the most appropriate strategies for targeted therapy. Functional evidence from our ORF screen analysis further highlights the biological relevance of *BRD4* deregulation in other cancer types, such as medulloblastoma, where targeted therapies against BRD4, including

(F) Quantification of cell confluency in OVSAHO-Cas9 cells following CRISPR-Cas9-mediated sgRNA cutting of *BRD4* regulatory regions or sgGFP control. Mean and standard deviation of three replicates, two-way ANOVA followed by Dunnett post-tests, controlling the Family-wise alpha threshold and confidence level. *p* values of the final time point are annotated.

(G) Landscape of *BRD4* dependency across ovarian cancers. Chronos scores (with mean and standard deviation) are shown for every ovarian cancer cell line included in the DepMap Public 23Q4 release. A score of −1 (median score for essential genes; red dashed line) is used as reference to indicate genetic dependency.

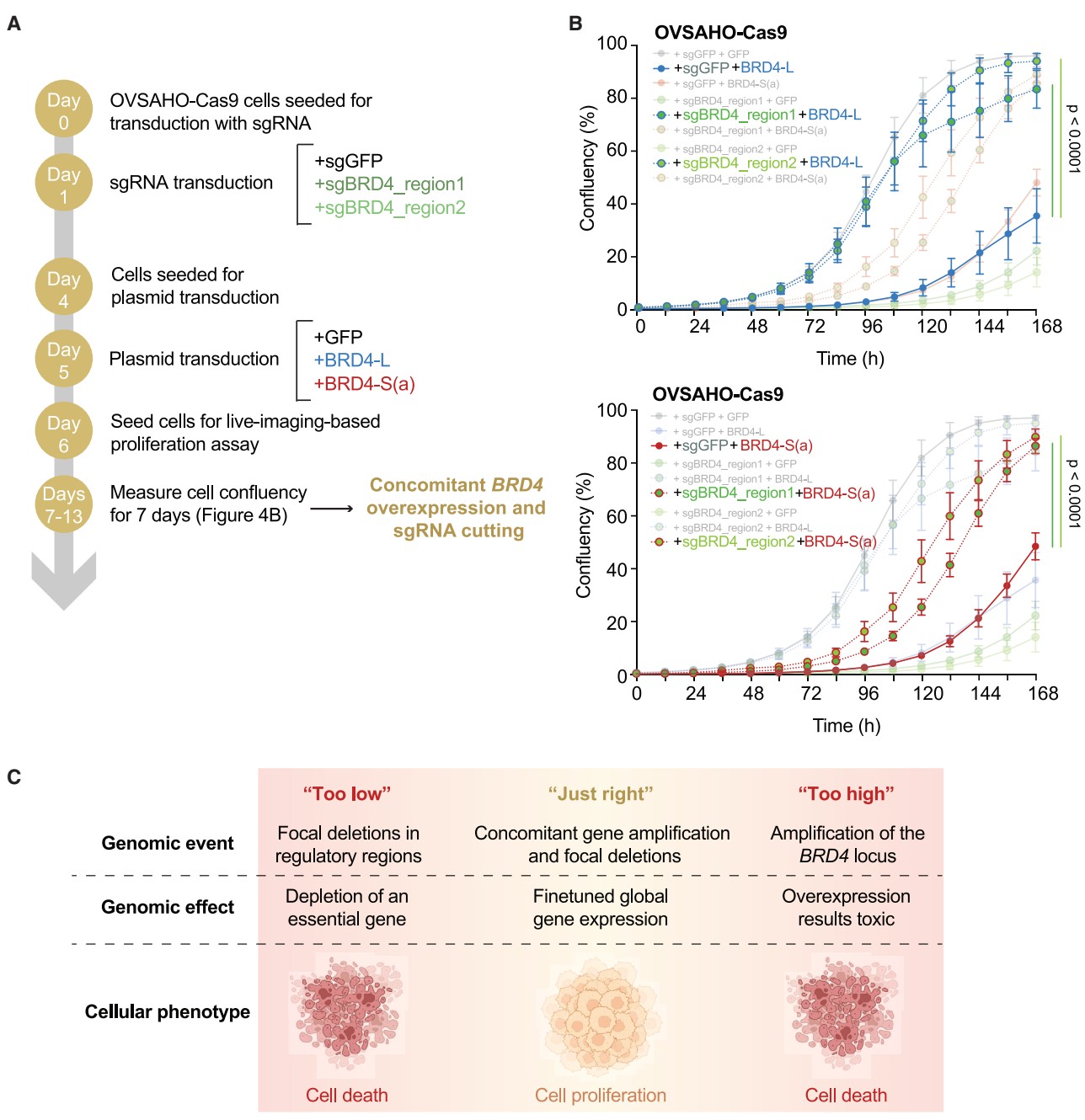

**Figure 5. Fine-tuned levels of *BRD4* global expression are required for sustained cellular proliferation**

(A) Experimental workflow for mimicking concomitant focal *BRD4* deletions occurring in *BRD4* overexpressing ovarian cancer cells.

(B) Rescue experiment by modulation of *BRD4* expression. Quantification of cell confluency after CRISPR-Cas9-mediated gene ablation was measured in OVSAHO-Cas9 cells over-expressing BRD4-L (top) or BRD4-S(a) isoforms (bottom). Mean and standard error from three replicates, two-way ANOVA followed by Dunnett post-tests, controlling the Family-wise alpha threshold and confidence level. *p* values of the final time point are annotated.

(C) Model of *BRD4* expression regulation. In ovarian cancers, too-low or too-high expression levels are detrimental to cellular fitness. In *BRD4*-amplified tumors, focal deletions serve as a mechanism to dampen global gene expression and rescue cellular proliferation.

**Cell Genomics**
**Article**

bromodomain and extra-terminal motif-bromodomain inhibitors, hold clinical promise.[23,33]

This is not the first case where genomic amplifications are toxic to cancer. In recent efforts, our groups and others have characterized the effect of collateral amplifications occurring across multiple tumor types, identifying genes that cause proliferation defects when overexpressed due to their close genomic proximity to an amplified oncogene. Such amplifications introduce genetic vulnerabilities and may present unexplored therapeutic avenues.[29,34–36] The present study provides evidence of *BRD4* focal deletions serving as a mechanism of gene-dosage compensation following gene amplification.

### Limitations of the study

In this study, we have highlighted the biological relevance of fine-tuning *BRD4* expression in ovarian cancer cells, as too-high or too-little levels of gene expression become detrimental to cancer cell growth. We observed interesting differences in cell growth inhibition when overexpressing BRD4-L and BRD4-S(a) isoforms in our engineered cell line models system, but our reagents and experimental design are limited in detecting the potential impact of functional BRD4-L degradation products in these proliferation phenotypes. Additionally, our cell line models express low endogenous levels of the BRD4(b) short isoform, limiting our interpretation of how amplification of this isoform impacts cancer cell fitness.

### RESOURCE AVAILABILITY

#### Lead contact
Further requests can be directed to Veronica Rendo (veronica.rendo@igp.uu.se).

#### Materials availability
BRD4-L and BRD4-S(a) overexpressing constructs, as well as CRISPR sgRNAs targeting regions 1 and 2 of *BRD4*, are available upon request to V.R. and P.B.

#### Data and code availability
Code is available at: https://github.com/walaj/brd4-analysis and at Zenodo via https://doi.org/10.5281/zenodo.14868313. PCAWG RNA sequencing expression data, copy-number data, and rearrangement data are also available at: https://data.broadinstitute.org/BRD4.

### ACKNOWLEDGMENTS

V.R. is supported by the Swedish Research Council (2022-01539), the Swedish Cancer Society (23 0622 JIA), and the Berth van Kantzows Foundation. J.W. was supported by an NIH T32CA009172 award from the National Cancer Institute (NCI) and is currently supported by a Wong Family Award in Translational Oncology. R.B. was supported by The Gray Matters Brain Cancer Foundation and the Brown Fund for Innovative Cancer Informatics. S.D. was supported by an NIH NRSA award from NCI (1F32CA261024) and is currently supported by an NIH National Institute of General Medical Sciences career development award (K99/R00), 1K99GM155595. The figures were created with BioRender.com.

### AUTHOR CONTRIBUTIONS

Conceptualization: J.W., R.B., P.B., and V.R. Methodology: J.B., P.B., and V.R. Formal analysis: J.W. S.D., S.W., S.S., R.B., P.B., and V.R. Investigation: J.W. S.D., S.W., O.S., J.B., S.S., P.B., and V.R. Resources: R.B. and P.B. Writing: J.W., S.D., S.W., R.B., P.B., and V.R. Visualization: J.W., S.D., S.W.,

and V.R. Supervision: R.B., P.B., and V.R. Funding acquisition: S.D., R.B., P.B., and V.R.

### DECLARATION OF INTERESTS

P.B. serves on paid advisory boards for DayOne Biopharmaceuticals and has served on a paid advisory board for QED Therapeutics. Her lab has received grant funding from Novartis Institute of Biomedical Research. J.W. has received consulting fees from Bullfrog AI.

### STAR★METHODS

Detailed methods are provided in the online version of this paper and include the following:

- KEY RESOURCES TABLE
- EXPERIMENTAL MODEL AND STUDY PARTICIPANT DETAILS
  - Cell line culture conditions
- METHOD DETAILS
  - Generation of lentiviral overexpression constructs
  - Generation of lentiviral Cas9 and single-guide RNA constructs
  - Quantification of gene expression by quantitative reverse transcription PCR (RT-qPCR)
  - Immunoblotting
  - Cas9 activity assay
  - CRISPR amplicon sequencing
  - Proliferation curves
- QUANTIFICATION AND STATISTICAL ANALYSIS
  - Structural variant analysis
  - BRD4 epigenomic analysis
  - Mutational timing analysis
  - BRD4 expression analysis
  - Linear modeling of BRD4 expression
  - GTEx expression analysis
  - Analysis of open reading frame (ORF) screens
  - Analysis of BRD4 dependency
  - Statistical analyses

### SUPPLEMENTAL INFORMATION

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

## STAR★METHODS

### KEY RESOURCES TABLE

| REAGENT or RESOURCE | SOURCE | IDENTIFIER |
|---|---|---|
| **Antibodies** | | |
| rabbit BRD4 antibody | ThermoFisher Scientific | Cat#702448; RRID: AB_2688043 |
| rabbit BRD4-L antibody | Bethyl Laboratories | Cat#A301-985A100; RRID: AB_1576498 |
| mouse SpCas9 antibody | Cell Signaling Technology | Cat#14697; RRID: AB_2750916 |
| rabbit GAPDH antibody | Cell Signaling Technology | Cat#2118; RRID: AB_561053 |
| goat anti-rabbit antibody (secondary antibody) | Cell Signaling Technology | Cat#7074; RRID: AB_2099233 |
| goat anti-mouse antibody (secondary antibody) | Cell Signaling Technology | Cat#7076; RRID: AB_330924 |
| **Bacterial and virus strains** | | |
| lentivirus pXPR_001 | Genetic perturbation platform, Broad Institute of MIT and Harvard | N/A |
| pXPR_BRD016 expressing human SpCas9 and sgRNA (see Table Oligonucleotide) | Genetic perturbation platform, Broad Institute of MIT and Harvard | N/A |
| pXPR_BRD111 expressing human SpCas9 and sgRNA (see Table Oligonucleotide) | Genetic perturbation platform, Broad Institute of MIT and Harvard | N/A |
| One Shot Stabl3 Chemically Competent *E. coli* | ThermoFisher Scientific | Cat#C737303 |
| **Chemicals, peptides, and recombinant proteins** | | |
| Fetal bovine serum (FBS) | ThermoFisher Scientific | Cat#16140071 |
| Penicillin-Streptomycin | ThermoFisher Scientific | Cat#15070063 |
| Advanced DMEM medium | ThermoFisher Scientific | Cat#12491015 |
| Puromycin | ThermoFisher Scientific | Cat#A1113803 |
| Blasticidin | ThermoFisher Scientific | Cat#A1113903 |
| Hygromycin B | ThermoFisher Scientific | Cat#10687010 |
| NuPAGE 4–12% Bis-Tris protein gel | ThermoFisher Scientific | Cat#NP0326 |
| NuPAGE MOPS Running Buffer | ThermoFisher Scientific | Cat#NP0001 |
| PVDF Transfer Stacks | ThermoFisher Scientific | Cat#IB24002 |
| SuperSignal West Pico PLUS Substrate | ThermoFisher Scientific | Cat#34580 |
| Phusion High-Fidelity PCR Master Mix with HF or GC Buffer | ThermoFisher Scientific | Cat#F532S |
| Advanced RPMI 1650 Medium | ThermoFisher Scientific | Cat#12633012 |
| Carbenicilin | ThermoFisher Scientific | Cat#10177012 |
| **Critical commercial assays** | | |
| QIAprep Spin Miniprep | Qiagen | Cat#27104 |
| Plasmid Plus Maxi Kits | Qiagen | Cat#12963 |
| Gateway LR Clonase II enzyme mix | Invitrogen | Cat#12535-029 |
| MycoAlert Mycoplasma detection kit | Lonza | Cat#LT07-318 |
| Lipofectamine 3000 Transfection Reagent | ThermoFisher Scientific | Cat#L3000001 |
| RNeasy Mini Kit | Qiagen | Cat#74004 |
| RNase-Free DNase Set | Qiagen | Cat#79254 |
| Maxima H Minus first-strand cDNA synthesis kit | ThermoFisher Scientific | Cat#K1652 |
| Maxima SYBR Green/ROX qPCR Master Mix | ThermoFisher Scientific | Cat#K0221 |
| Pierce BCA Protein Assay | ThermoFisher Scientific | Cat#23225 |
| **Experimental models: Cell lines** | | |
| Human: OVSAHO (female sex) | Sigma-Aldrich | Cat#SCC294 |
| Human: HEK-293T (female sex) | ATCC | Cat#CRL-11268 |

*Continued*

| REAGENT or RESOURCE | SOURCE | IDENTIFIER |
|---|---|---|
| **Oligonucleotides** | | |
| sgBRD4_region1 Forward | Integrated DNA Technologies | 5′-CACCGTCTACCACACTGTAATGCTG-3′ |
| sgBRD4_region1 Reverse | Integrated DNA Technologies | 5′-AAACCAGCATTACAGTGTGGTAGAC-3′ |
| sgBRD4_region2 Forward | Integrated DNA Technologies | 5′-CACCGCGTAGTCTTGAGTAAAAGGG-3′ |
| sgBRD4_region2 Reverse | Integrated DNA Technologies | 5′-AAACCCCTTTTACTCAAGACTACGC-3′ |
| sgGFP Forward | Integrated DNA Technologies | 5′-CACCGGAGCTGGACGGCGACGTAAA-3′ |
| sgGFP Reverse | Integrated DNA Technologies | 5′-AAACTTTACGTCGCCGTCCAGCTCA-3′ |
| **Recombinant DNA** | | |
| psPAX2 | Addgene | Cat#12260 |
| pVSV-G | Addgene | Cat#138479 |
| pDONR223 encoding BRD4-L | Genetic perturbation platform, Broad Institute of MIT and Harvard | Clone ID: ccsbBroadEn_11738; NM_058243.3 |
| pDONR223 encoding BRD4-S(a) | Genetic perturbation platform, Broad Institute of MIT and Harvard | Clone ID: ccsbBroadEn_11738; NM_014299.2) |
| pDONR223 encoding GFP | Genetic perturbation platform, Broad Institute of MIT and Harvard | Clone ID: BRDN0000464762 |
| pLX_307 | Genetic perturbation platform, Broad Institute of MIT and Harvard | N/A |
| **Software and algorithms** | | |
| CRISPResso2 | Clement et al.[37] | Clement et al.[37] |
| ImageJ | Wayne Rasband and contributors, National Institutes of Health, USA. Java 1.8.0_251 (64-bit) | N/A |
| PRISM | GraphPad | Version 10 |
| EnhancedVolcano (R package) | https://github.com/kevinblighe/EnhancedVolcano | 1.20 |
| limma (R package) | Ritchie et al.[38] | 3.58.1 |
| gTrack (R package) | https://github.com/mskilab-org/gTrack | N/A |
| UCSC Table Browser | https://genome.ucsc.edu/cgi-bin/hgTables | N/A |
| Primary analysis notebook | https://github.com/walaj/brd4-analysis/ | https://doi.org/10.5281/zenodo.14868313 |
| BioRender | https://BioRender.com | N/A |
| **Other** | | |
| 2720 Thermal Cycler | Applied Biosystems | N/A |
| Incucyte Live-Cell Analysis system | Sartorius | N/A |
| CRISPR amplicon sequencing | CCIB DNA Core - Massachusetts General Hospital | N/A |
| LSRFortessa Cell Analyzer | BD Biosciences | N/A |
| iBlot 2 | Invitrogen | N/A |
| ImageQuant LAS 4000 imager | GE Healthcare LIfe Sciences | N/A |

## EXPERIMENTAL MODEL AND STUDY PARTICIPANT DETAILS

### Cell line culture conditions

OVSAHO human high-grade serous ovarian cancer cells were commercially acquired (cat. no. SCC294, Sigma-Aldrich) and cultured in Advanced RPMI 1640 Medium (cat. no. 12633012, ThermoFisher Scientific) supplemented with 10% heat inactivated Fetal Bovine Serum (FBS) and 1% Pen/Strep (cat. no. 16140071 and 15070063, ThermoFisher Scientific). Human HEK-293T were commercially acquired (cat. no. CRL-11268, ATCC) and grown in Advanced DMEM medium (cat. 330 no. 12491015, ThermoFisher Scientific) supplemented with 10% FBS and 1% Pen/Strep. All cell lines were incubated at 37°C with 5% $CO_2$, fingerprinted, and frequently checked for mycoplasma contamination using the MycoAlert Mycoplasma Detection Kit (cat. no. LT07-318, Lonza).

## METHOD DETAILS

### Generation of lentiviral overexpression constructs

pDONR223 vectors encoding the BRD4-L long (clone ID: ccsbBroadEn_11738; NM_058243.3) and BRD4-S(a) short (clone ID: ccsbBroadEn_11738; NM_014299.2) isoform sequences, or a GFP control (clone ID: BRDN0000464762), were acquired from the Genomic Perturbation Platform (GPP) at the Broad Institute of MIT and Harvard. Next, each coding sequence was cloned into the pLX_307 lentiviral vector (GPP, Broad Institute of MIT and Harvard), driven by a EF1a promoter and under puromycin selection, using the Gateway LR Clonase II enzyme mix (cat. no. 12535-029, Invitrogen) according to the manufacturer's protocol. The resulting products from the LR recombination reaction were then transformed into One Shot Stabl3 Chemically Competent *E. coli* (cat. no. C737303, ThermoFisher Scientific) at 37°C in LB microbial growth medium containing 100 μg/mL carbenicillin. Plasmid DNA was extracted from single colonies using QIAprep Spin Miniprep (cat. no. 27104, Qiagen) and Plasmid Plus Maxi Kits (cat. no. 12963, Qiagen). Lentiviral transduction was performed by co-transfection of HEK-293T cells with 10 μg of pLX_307 vector (BRD4 or GFP control), 1 μg psPAX2 (plasmid #12260, Addgene) and 1 μg pVSV-G (plasmid #138479, Addgene) viral packaging plasmids using Lipofectamine 3000 Transfection Reagent (cat. no. L3000001, ThermoFisher Scientific). After 48 h, virus harvesting was conducted using a 0.45 μm syringe. For cell transduction, 400 μL of virus were added to OVSAHO cells seeded at a density of 2 million cells per well in a 12-well plate. After an overnight incubation at 37°C, cells were selected for 5 days with 1 μg/mL puromycin (cat. no. A1113803, ThermoFisher Scientific).

### Generation of lentiviral Cas9 and single-guide RNA constructs

The lentiviral vectors pXPR_BRD111 and pXPR_BRD016 expressing human SpCas9 and a single-guide RNA (sgRNA) of interest, respectively, were acquired from the GPP Platform (Broad Institute of MIT and Harvard). While SpCas9 lentiviral expression is under blasticidin selection, expression of the sgRNA of interest is driven by a human U6 promoter in a construct under hygromycin selection. The sgRNAs against BRD4_region1 and BRD4_region2 were cloned into the pXPR_BRD016 backbone following the lentiCRISPRv2 one vector system protocol described by the Zhang laboratory.[39,40] The oligos designed against sgBRD4_region1 were: forward 5′-CACCGTCTACCACACTGTAATGCTG-3′, and reverse 5′-AAACCAGCATTACAGTGTGGTAGAC-3′. The oligos designed against sgBRD4_region2 were: forward 5′-CACCGCGTAGTCTTGAGTAAAAGGG-3′, and reverse 5′-AAACCCCTTTTTACTCAAGACTACGC-3′. As a control, we cloned a sgRNA targeting GFP ("sgGFP") using oligos forward 5′-CACCGGAGCTGGACGGCGACGTAAA-3′ and reverse 5′-AAACTTTACGTCGCCGTCCAGCTCA-3′. Lentiviral production and transduction were performed in HEK-293T and OVSAHO cells as described above. Selection of OVSAHO Cas9 over-expressing cells was conducted for 7 days with 10 μg/mL blasticidin (cat. no. A1113903, ThermoFisher Scientific). After transduction with sgRNA-expressing constructs, a second round of selection was performed for 10 days with 100 μg/mL hygromycin B (cat. no. 10687010, ThermoFisher Scientific).

### Quantification of gene expression by quantitative reverse transcription PCR (RT-qPCR)

RNA was first extracted from one million cells using the RNeasy Mini Kit (cat. no. 74004, Qiagen) using an on-column DNase digestion (cat. no. 79254, Qiagen). Next, cDNA conversion was conducted from 2 μg purified RNA using the Maxima H Minus first-strand cDNA synthesis kit (cat. no. K1652, ThermoFisher Scientific). Quantitative PCR analysis was done on 500 ng cDNA mixed with 1X Maxima SYBR Green/ROX qPCR Master Mix (cat. no. K0221, ThermoFisher Scientific) and 0.3 μM primers. For the detection of *BRD4* isoform transcripts, the following primers were used: BRD4-long forward 5′-CCGGCTCCTCCAAGATGAAG-'3′ and reverse 5′-ATGGTGCTTCTTCTGCTCCC-3′; BRD4-short forward 5′-CCTCAAGCTGAGAAAGTTGATGTG-'3′ and reverse 5′-AGGACCTGTTTCGGAGTCTT-3′. As an endogenous control, primers forward 5′-CCGAAAGTTGCCTTTTATGG-3′ and reverse 5′-TCATCATCCATGGTGAGCTG-3′ targeting *ACTB* were used. Samples were run in a StepOne thermocycler according to the following protocol: 40 cycles of 95°C for 15 s and 60°C for 1 min, melt curve stage at 95°C for 15 s, and 60°C for 1 min. Quantification of mRNA expression relative to control was performed by the ΔΔCT method.

### Immunoblotting

Two million cells were pelleted and lysed in RIPA buffer, containing 20 mM Tris-HCl at pH 7.5, 150 mM NaCl, 1 mM Na$_2$EDTA, 1 mM EGTA, 1% NP-40, 1% sodium deoxycholate, 2.5 mM sodium pyrophosphate, 1 mM beta-glycerophosphate, 1 mM Na$_3$VO$_4$, and 1 μg/mL leupeptin. After a 30 min incubation on ice, samples were centrifuged at 13,000 rpm for 10 min at 4°C. Protein concentrations were quantified using Pierce BCA Protein Assay (cat. no. 23225, ThermoFisher Scientific). Next, 30 μg of each sample was loaded on a NuPAGE 4–12% Bis-Tris protein gel (cat. no. NP0326, ThermoFisher Scientific) and separated using NuPAGE MOPS Running Buffer (cat.no. NP0001, ThermoFisher Scientific) for 2 h at 120 V. A dry gel transfer was completed at 30 V for 6 min using PVDF Transfer Stacks (cat. no. IB24002, ThermoFisher Scientific) in an iBlot 2 instrument (Invitrogen). After blocking with 5% dry milk in TBS-T for 30 min, the membrane was incubated at 4°C overnight with primary antibodies against rabbit BRD4 (cat. no. 702448, ThermoFisher Scientific), rabbit BRD4-L (cat. no. A301-985A100, Bethyl Laboratories), mouse SpCas9 (cat. no. 14697, Cell Signaling Technology), or rabbit GAPDH (cat. No. 2118, Cell Signaling Technology). Next, the membrane was washed in TBS-T and incubated for 1h with goat anti-rabbit (cat. no. 7074, Cell Signaling Technology) or goat anti-mouse (cat. no. 7076, Cell Signaling Technology) secondary antibodies. Chemiluminescent signal was detected using SuperSignal West Pico PLUS Substrate (cat. no. 34580, ThermoFisher Scientific) in an ImageQuant LAS 4000 imager (GE Healthcare Life Sciences).

**Cell Genomics**
**Article**

## Cas9 activity assay

One million OVSAHO or OVSAHO-Cas9 expressing cells were transduced with the lentivirus pXPR_011 (GPP, Broad Institute of MIT and Harvard), which co-expresses a sgRNA against EGFP and EGFP as a target under puromycin selection. Following infection with 480 μL virus, cells were spun for 2 h at 1,000 $g$ at 30°C, and cell media was replaced after 6 h. The next day, 1 μg/mL puromycin was added. To quantify Cas9 activity, the percentage of GFP-negative cells (where Cas9-mediated cutting of EGFP has occurred) was quantified in OVSAHO and OVSAHO-Cas9 cells by flow cytometry using a BD LSRFortessa Cell Analyzer (BD Biosciences) using a 488 nm laser. Non-expressing Cas9 OVSAHO cells appear green and provide a baseline reference for determining Cas9 activity in OVSAHO-Cas9 cells.

## CRISPR amplicon sequencing

To confirm CRISPR-mediated cutting of sgBRD4_region1 and sgBRD4_region2, we first designed primers spanning the regions of interest in *BRD4*'s intron 1. For "region 1", we used forward 5′-AGATCCTTTTGGCTCCCTGT-3′ and reverse 5′-GGGAGAGAG AGGTTGCAGTG-3′ primers. For "region 2" we used forward 5′-TTGCTTGAAGATGGGAAACC-3′ and reverse 5′-ACGGTAGGG AACTTGACAGC-3′ primers. Next, we amplified each region by PCR using 1× Phusion HF Buffer (Thermo Scientific), 0.2 mM dNTPs, 0.2 μM forward, and 0.2 μM reverse primers, 0.02 U μL−1 Phusion DNA Polymerase (Thermo Scientific) and 6 ng of genomic DNA. The PCR was performed in a 2720 Thermal Cycler (Applied Biosystems) following the protocol: 95°C for 30 sec, 30 cycles of i) 95°C for 10 sec, ii) 58°C for 15 sec, and iii) 72°C for 20 sec, and 72°C for 10 min. CRISPR amplicon sequencing was performed by next-generation sequencing at the CCIB DNA Core (Massachusetts General Hospital). All reads were aligned to NCBI's *BRD4*'s reference sequence and analyzed with CRISPResso2[37] to obtain editing frequencies and indel size distribution for each targeted region.

## Proliferation curves

Cells were seeded at a density of 2,000 cells per well in 96-well plates. The next day, live-cell imaging was conducted in an Incucyte Live-Cell Analysis system (Sartorius) and images were captured for each well every 12 h with a 10x objective. Custom-defined masks were then applied to each image to identify cell area and determine confluency as a surrogate of cell growth.

## QUANTIFICATION AND STATISTICAL ANALYSIS

### Structural variant analysis

We first analyzed PCAWG samples with amplifications and somatic SV breakpoints overlapping the *BRD4* locus using GISTIC gene-level thresholded copy number data and consensus-called SV and SCNA callsets from the ICGC Data Portal's DCC Data Release [https://dcc.icgc.org/releases/PCAWG/consensus_cnv/GISTIC_analysis]. We identified cases with focal *BRD4* deletions as those cases with an SCNA breakpoint within (or within 50 Kbp) of the *BRD4* locus leading to a relative copy-loss within *BRD4*, and with a total copy-loss SCNA event size of <100 Kbp. An additional 6 cases of *BRD4* focal deletions were identified by manual review of tumor/normal read depth plots, with SVs (rearrangements) annotated, for all breast, ovarian and endometrial cancers containing *BRD4* breakpoints. Read depth ratios were calculated for 200 bp intervals from raw read depth signal and plotted with the gTrack package (https://github.com/mskilab-org/gTrack) in R (4.3.2). These cases were deemed to have a true focal deletion if there was a visible decrease in read-depth signal with drop-offs corresponding to the breakpoints from the SV data, and if those breakpoint orientations were compatible with a copy-loss (outgoing orientation from the deletion) consistent with a deletion.

Total copy-number for any given locus (e.g., *CCNE1*) was taken as the mean total copy-number across that gene, using hg19 genomic coordinates obtained from Ensembl[41] human genome build hg19. For calculating copy-number normalized gene expression of *BRD4*, we took the copy-number to be that of the neighboring gene *NOTCH3*, since the copy-number within *BRD4* for *BRD4* focal deletion samples was variable by definition. We performed significance testing using a Wilcoxon rank-sum test with the R stats package. We identified tumors containing an amplification spanning the *BRD4* locus by intersecting the SCNA data for events with total copy-number >2 with the *BRD4* locus. This was determined irrespective of whether there was a concomitant *BRD4* focal deletion occurring on top of an amplification. We performed a Fisher's exact test with the R stats package to determine if *BRD4* focal deletions were enriched in amplified *BRD4* loci, using a *p*-value significance cutoff of 0.05.

### BRD4 epigenomic analysis

H3K4me3 peaks from the ENCODE consortium with the following UCSC accession numbers were downloaded from UCSC Table Browser: wgEncodeEH000967 (MCF-7) and wgEncodeEH000954 (HMEC). H3K4me3 data were plotted at the *BRD4* locus as shown in Figure 1A and qualitatively compared against the mean read depth signal for the 22 cases of *BRD4* focal deletions.

### Mutational timing analysis

To assess whether selection favored a particular order to the events, we phased allelic copy number in samples with both amplifications and focal deletions in *BRD4*. Of the 19 samples meeting these criteria, 13 had the discernible allelic copy number information needed to perform this analysis. In each sample, if one allele demonstrated an amplification and loss of heterozygosity, we surmised that a deletion happened first. Conversely, if one allele exhibited an amplification and a deletion resulting in a non-zero copy number, we inferred the amplification occurred first. In the remaining six samples where the events occurred on different alleles, the deletion

was too small to register an effect on allelic copy number, or the sample exhibited uniparental disomy, we deemed the sample "ambiguous" with respect to event timing. Allelic copy number could not be ascertained for these cases due to the small deletion widths. An exception is one sample (BRCA-UK::CGP_donor_1199131) with a uniparental disomy event that was evidently preceded by an amplification.

## BRD4 expression analysis

Gene level and transcript level RNA expression data was downloaded from the PCAWG data portal as above. RNA expression values were used as previously described[42] and without further modification. BRD4-L corresponded to Ensembl transcript ENST00000263377. The primary BRD4-S transcripts corresponded to Ensembl transcript ENST00000371835 (for dominant BRD4-S(a)) and to transcript ENST00000360016.

Differential expression analysis was performed using the limma package (v. 3.58.1[38]) in R and the voom method for linear modeling,[43] comparing expression from tumors with a *BRD4* focal deletion (*n* = 18 with tumor RNA-seq data) against tumors without a *BRD4* focal deletion (*n* = 1266). Expression data were normalized with the calcNormFactors function, followed by variance-stabilizing transformation using the voom method. Linear models were fitted to each gene, adjusting for tumor type, and empirical Bayes moderation was applied to improve statistical power. Expression analysis was restricted to just protein coding genes on autosomal chromosomes. We performed differential expression analysis comparing two model matrix approaches. The first modeled just the *BRD4* focal deletion status. The second also included tumor type as obtained from the ICGC Data Coordinating Center (DCC) project code (e.g., BRCA tumors consisted of all tumors from projects: BRCA-EU, BRCA-US and BRCA-UK). Multiple hypothesis correction was performed using Bonferroni correction. The volcano plot was generated using the EnhancedVolcano package (1.20.0).

## Linear modeling of BRD4 expression

Linear regression modeling was used to analyze the relationship between *BRD4* expression and absolute copy number, *BRD4* focal deletion status, and tumor type (BRCA, OV, UCEC, and Other). Tumor types "other" (not breast, ovarian or endometrial) was set as the reference level. Where multiple tumor samples were profiled in the same patient, expression data was calculated as the mean of the per-patient levels. Analyses and model fitting were conducted using base R packages, with regression coefficients and 95% confidence interval estimates extracted for downstream interpretation and visualization.

## GTEx expression analysis

For the analysis of *BRD4* isoform expression across normal tissues, we downloaded transcript per million (TPMs) measurements from the GTEx Analysis V8 RNA-sequencing data [https://gtexportal.org/home/downloads/adult-gtex/bulk_tissue_expression], and compared transcript expression levels across available tissue types using paired Wilcoxon signed-rank tests with a Bonferroni correction.

## Analysis of open reading frame (ORF) screens

The analysis of ORF screen data was performed as previously described.[29] Briefly, we looked at data from 16 independent screens conducted at the Broad Institute of MIT and Harvard, where cells were infected with the ORFeome pLX_317 barcoded library, which contains 16,100 barcoded ORFs over-expressing 12,753 genes. From these, we compared ORF abundance between early and late timepoints for BRD4-L (TRCN0000477053; NM_058243.2) and BRD4-S(a) (TRCN0000491254; NM_014299.2) isoforms. A negative log2-fold change in cell growth following ORF expression as interpreted as an indication of gene toxicity.

## Analysis of BRD4 dependency

CRISPR dependency scores (Chronos) were downloaded from the DepMap Public 23Q4 release (https://forum.depmap.org/t/announcing-the-23q4-release/2929) for all available ovarian cancer cell lines (*n* = 59). A Chronos score threshold of −1 was used to call *BRD4* a genetic dependency in each cell line.

## Statistical analyses

Analyses were performed in R (4.3.2) and PRISM 10. Replicate numbers and statistical tests are notated on figures and in figure legends when relevant. Generally, when comparing two groups, for example log-fold change of BRD4-L vs. BRD4-S isoforms in our ORF screen, we used Wilcoxon rank sum tests, as noted in figure legends. When across multiple pairs of groups, we corrected *p*-values for multiple hypotheses using the Bonferroni correction. When comparing the effect of more than two treatments, for example knocking out and overexpressing BRD4-L and BRD4-S, we used a two-way ANOVA followed by Dunnett post-tests to compare every mean to a GFP + sgGFP control mean. In all cases, at least three replicates are included.

