## [Document S2. Transparent peer review records for Wala et al. · Cell Genomics]

Recurrent breakpoints in the BRD4 locus reduce toxicity associated with gene amplification

Author list: Jeremiah Wala, Simona Dalin, Sophie Webster, Ofer Shapira, John Busanovich, Shahab Sarmashghi, Rameen Beroukhim, Pratiti Bandopadhyay, Veronica Rendo

Summary

Initial submission: Received : August 22nd 2024

Scientific editor: Laura Zahn

First round of review: Number of reviewers: 2
Revision invited : October 21st 2024
Revision received : December 9th 2024

Second round of review: Number of reviewers: 2
Accepted : 21st February 2025

Data freely available: Yes

Code freely available: Yes

This transparent peer review record is not systematically proofread, type-set, or edited. Special characters, formatting, and equations may fail to render properly. Standard procedural text within the editor's letters has been deleted for the sake of brevity, but all official correspondence specific to the manuscript has been preserved.

Referees' reports, first round of review

Reviewer 1

MS#: CELL-GENOMICS-D-24-00463

Authors: Wala, Dalin, Webster, Shapira, Busanovich, Beroukhim, Bandopadhyay, and Rendo

In this manuscript, Wala et al. analyzed focal gene deletions in the BRD4 locus, focusing on p19 somatic copy-number alterations (SCNA). The authors found recurrent focal BRD4 gene deletions seem to diminish the potentially toxic effects of p19 SCNA on mostly breast and ovarian cancer cell proliferation. Focal deletion-reduced expression of both BRD4-long (BRD4-L) and -short (BRD4-S) isoforms contribute to this homeotic maintenance of cancer cell growth. Overall, the finding of BRD4 gene-amplified tumors undergoing focal rearrangements to diminish BRD4 overexpression, thus maintaining proper gene dosage of a critical growth regulator, is conceptually interesting. However, some key points need to be further addressed:

1. The term "oncogenic BRD4 gene" is confusing and not in line with the critical role of BRD4 in maintaining normal cell growth and gene function. This is particularly relevant when both tumor-suppressive and oncogenic functions of BRD4 have been attributed to distinct BRD4-L and BRD4-S isoforms, respectively, in breast cancer.
2. The protein levels of BRD4 isoforms could not be entirely deduced from their transcript levels. While the RNA levels of BRD4-L and BRD4-S could be analyzed by their unique terminal exons, the protein levels of these two protein isoforms need to be convincingly demonstrated by using isoform-specific antibodies. This is particularly important considering a predominant long form degradation product coinciding with the short form is often detected in many cell extracts. Without proper controls, data interpretation could sometimes be misleading.
3. Rearranged BRD4 locus could also affect the expression of another BRD4 short isoform, termed short form b - BRD4-S(b). It would be nice if the authors could also analyze whether recurrent focal BRD4 deletions detected in various cancer types may also affect BRD4-S(b) transcript levels.

4. BRD4 overexpression has been detected in many cancer types and correlates with oncogenic potential. This appears to be contradictory to the view conveyed in the current study. Since dose response is critical in biological readout, the authors may need to carefully define what levels of "overexpression" refer to in their context.

5. Immunoblotting performed using lysates prepared in RIPA buffer containing 150 nM NaCl may not be right.

Reviewer 2

In the manuscript titled "Recurrent breakpoints in the BRD4 locus reduce toxicity associated with gene amplification", the authors using genomic tools and OVSAHO ovarian cancer cells experiments identified and elucidated a 'Goldilocks' model of BRD4 gene regulation, where too high or too low levels of expression have a negative effect on cellular fitness. In BRD4-amplified tumors, focal deletions in gene regulatory regions serve as a mechanism to decrease and restore gene expression to levels that are tolerated and beneficial for proliferation. This represents a novel mechanism of gene-dosage compensation in human cancers.

It's a very interesting study that provides the first experimental evidence for a novel mechanism by which genetic alterations drive cancer. However, some points should be addressed:

1. The amplification curves of the +GFP group in Figure 3F and Figure 4F are very similar, please check whether the same set of data was used. In addition, the amplification curves of the +BRD4-long and +BRD4-short groups in Figure 3F and Figure 5B are also very similar, please check whether the same set of data was used as well.

2. In Figure 5A, the authors first simulated focal deletions in the BRD4 regulatory region by CRISPR-Cas9 technology in OVSAHO cells, and then transfected BRD4-long and BRD4-short, and found that these focal deletions could rescue ovarian carcinoma cells from the toxic effects associated with gene overexpression. I would be also very interested if the order of process was changed, with transfection of BRD4-long and BRD4-short first to simulate the overexpression of BRD4-long and BRD4-short, and then the focal deletion of the BRD4 regulatory region was simulated by CRISPR-Cas9 technology. This order seems to shed more

light on whether focal deletions protect ovarian cancer cells from the toxic effects of gene overexpression.

3. The paper uses a variety of statistical methods, just some of which are described, please add a general description of statistical methods for others.

4. Please note the individual writing errors.

(1) In line 327, "RPMI 1460 Medium" should be "RPMI 1640 Medium".

(2) In line 329, "1% Pen Strep" should be "1% Pen/Strep".

(3) In line 330, "Advanced DMEM (cat. 330 no. 12491015, ThermoFisher Scientific) medium supplemented with 10% FBS" should be changed to "Advanced DMEM medium (cat. 330 no. 12491015, ThermoFisher Scientific) supplemented with 10% FBS". In addition, please check if 1% Pen/Strep is also added here.

(4) In line 345, "ug/mL" should be "µg/mL".

Authors' response to the first round of review

Reviewer #1:

MS#: CELL-GENOMICS-D-24-00463

Authors: Wala, Dalin, Webster, Shapira, Busanovich, Beroukhim, Bandopadhyay, and Rendo

In this manuscript, Wala et al. analyzed focal gene deletions in the BRD4 locus, focusing on p19 somatic copy-number alterations (SCNA). The authors found recurrent focal BRD4 gene deletions seem to diminish the potentially toxic effects of p19 SCNA on mostly breast and ovarian cancer cell proliferation. Focal deletion-reduced expression of both BRD4-long (BRD4-L) and -short (BRD4-S) isoforms contribute to this homeotic maintenance of cancer cell growth. Overall, the finding of BRD4 gene-amplified tumors undergoing focal rearrangements to diminish BRD4 overexpression, thus maintaining proper gene dosage of a critical growth regulator, is conceptually interesting. However, some key points need to be further addressed:

1. The term "oncogenic BRD4 gene" is confusing and not in line with the critical role of BRD4 in maintaining normal cell growth and gene function. This is particularly relevant when both tumor-suppressive and oncogenic functions of BRD4 have been attributed to distinct BRD4-L and BRD4-S isoforms, respectively, in breast cancer.

We agree with the reviewer that assigning the term "oncogenic" to describe BRD4's cellular function may be misleading, in particular since the BRD4-long isoform has been attributed a tumor suppressive role while BRD4-short is deemed oncogenic. We have now removed the term "oncogenic BRD4" from the abstract and main text (lines 21 and 50) and have clarified that different BRD4 isoforms have been attributed with distinct roles in the discussion (lines 256 and 257).

2. The protein levels of BRD4 isoforms could not be entirely deduced from their transcript levels. While the RNA levels of BRD4-L and BRD4-S could be analyzed by their unique terminal exons, the protein levels of these two protein isoforms need to be convincingly demonstrated by using isoform-specific antibodies. This is particularly important considering a predominant long form degradation product coinciding with the short form is often detected in many cell extracts. Without proper controls, data interpretation could sometimes be misleading.

Given the use of isogenic cell line models, we know that the only difference between parental OVSAHO cells and BRD4-short/-long overexpressing cells is the sequence of the corresponding isoform we have transduced cells with. In this sense, even if parental OVSAHO cells express a BRD4-long isoform and a degradation product that coincides with BRD4-short, and even if this degradation product has a functional role, we do not expect this to account for the differences in growth phenotypes observed between control cells and BRD4-short cells throughout our experiments.

We do agree with the reviewer that the use of isoform-specific antibodies can provide a more precise estimation of protein isoform expression. To this end, we performed an immunoblot in OVSAHO cells overexpressing GFP control, BRD4-short or BRD4-long isoforms with a polyclonal BRD4 antibody (Bethyl Laboratories, Cat. no. A301-985A100) that specifically binds to a region between residues 1312 and 1362 of the human BRD4 protein. These 50 amino acids are located in the C-terminal region of BRD4, and are present specifically in the BRD4-long but not the BRD4-short isoform. In this experiment, we detected BRD4-long protein expression in all samples, with increases in protein expression only being present in

BRD4-long overexpressing cells (Reviewer Figure 1).

Determination of BRD4-short protein with isoform-specific antibodies is challenging and less reliable, as a multiple sequence alignment (Clustal O) reveals that the coding sequences for these two isoforms completely overlap:

CLUSTAL O(1.2.4) multiple sequence alignment

Long - BRD4-201 - UniProt O60885-1

Short - BRD4-203 - UniProt O60885-1

```

long  MSAESGPGTRLRNLPMVDGLETSMSTTQAQAQPANAASTNPPPPETSNPKNPKRQT 60
short MSAESGPGTRLRNLPMVDGLETSMSTTQAQAQPANAASTNPPPPETSNPKNPKRQT 60
*****
  
```

```

long  NQLQYLLRVVLKTLWKHQFAWPFQQPVDVAVKLNLPDYYKIIKTPMDMGTIKKRENNYYW 120
short NQLQYLLRVVLKTLWKHQFAWPFQQPVDVAVKLNLPDYYKIIKTPMDMGTIKKRENNYYW 120
*****
  
```

```

long  NAQECIQDFNTMFTNICYIYNKPGDDIVLMAEALEKFLQKINELPTEETEIMIVQAKGRG 180
short NAQECIQDFNTMFTNICYIYNKPGDDIVLMAEALEKFLQKINELPTEETEIMIVQAKGRG 180
*****
  
```

```

long  RGRKETGTAKPGVSTVPNTTQASTPPQTTPQPNPPVQATPHFPFAVTPDLIVQTPVMT 240
short RGRKETGTAKPGVSTVPNTTQASTPPQTTPQPNPPVQATPHFPFAVTPDLIVQTPVMT 240
*****
  
```

```

long  VPPQPLQTPPPVPPQPAPPAPQPVQSHPPIIAATPQPVKTKKGVKRAKADTTTPTTI 300
  
```

short	VPPQPLQTPPPVPPQPQPPAPAPQPVQSHPPIIAATPQPVKTKKGVKRKADTTTPTTI	300

long	DPIHEPPSLPPEPKTTKLGQRRESSRPVKPPKKDVPDSQQHPAPEKSSKVSEQLKCCSGI	360
short	DPIHEPPSLPPEPKTTKLGQRRESSRPVKPPKKDVPDSQQHPAPEKSSKVSEQLKCCSGI	360

long	LKEMFAKKHAAYAWPFYKPDVEALGLHDYCDIHKHPMDMSTIKSKLEAREYRDAQEFGA	420
short	LKEMFAKKHAAYAWPFYKPDVEALGLHDYCDIHKHPMDMSTIKSKLEAREYRDAQEFGA	420

long	DVRLMFSNCKYKYNPPDHEVWAMARKLQDVFEMRFAKMPDEPEEPVAVSSPAVPPPTKV	480
short	DVRLMFSNCKYKYNPPDHEVWAMARKLQDVFEMRFAKMPDEPEEPVAVSSPAVPPPTKV	480

long	APSSSSSSSSSSSSSSDSDSDDSEEEAQRRLAELQEQLKAVHEQLAALSQPQQNKPKKKE	540
short	APSSSSSSSSSSSSSSDSDSDDSEEEAQRRLAELQEQLKAVHEQLAALSQPQQNKPKKKE	540

long	KDKKEKKEKHKRKEEVEENKSKAKEPPPCKTKKNSSNSNVSKKEPAMKSKPPPTYE	600
short	KDKKEKKEKHKRKEEVEENKSKAKEPPPCKTKKNSSNSNVSKKEPAMKSKPPPTYE	600

long	SEEDKCKPMSYEEKRQLSLDINKLPGEKLRVVIHQSRPESLKNPNPDEIEIDFETLK	660
short	SEEDKCKPMSYEEKRQLSLDINKLPGEKLRVVIHQSRPESLKNPNPDEIEIDFETLK	660

long	PSTLRELERYVTSCLRKKRKPQAEKVDVIAGSSKMGFSSSESSSESSSSSSDSEDSETE	720
short	PSTLRELERYVTSCLRKKRKPQAEKVDVIAGSSKMGFSSSESSSESSSSSSDSEDSETG	720

long	MAPSKKKKGHPGREQKHHHHHHQQMQQAPAPVPQQPPPPPPQPPPPPPQQQQPPPPP	780
short	PA-----	722
*		
long	PPSPMPQQAAPAMKSSPPPIATQVPVLEPQLPGSVFDPIGHFTQPILHLPQPELPPHP	840
short	-----	722
long	QPPEHSTPPHLNQHAVVSPALHNALPQQSRPSNRRAALPPKPARPPAVSPALQTPTLL	900
short	-----	722
long	PQPPMAQPPVLLLEDEEPPAPLTSMQMLYLQQLQKVPPTPLLPVSVKVSQPPPPPLPP	960
short	-----	722
long	PPHPSVQQQLQQQPPPPPPQPPPPQPPHQPVRVHLQPMQFSTHIQQPPPPQGGQPP	1020
short	-----	722

```

long  HPPPGQQPPPPQPAKPPQVIQHHSRHHKSDPYSTGHLREAPSPLMIHSPQMSQFQSLT 1080
short ----- 722

long  HQSPPQQNVQPKKQELRAASVVQPQLVWVKEEKIHSPIIRSEPFSPSLRPEPPKHPESI 1140
short ----- 722

long  KAPVHLPQRPEMKPVDVGRPVIRPPEQNAPPPGAPDKDKQKQEPKTPVAPKDLKIKNMG 1200

short ----- 722

long  SWASLVQKHPTTSSSTAKSSDSFEQFRAAREKEEREKALKAQAEHAEKEKERLRQERM 1260
short ----- 722

long  RSREDEDALQARRAHEEARRRQEQQQQRQEQQQQQQQAAAATAATPQAQSSQPQSM 1320
short ----- 722

long  LDQQRELARKREQERRRREAMAATIDMNFQSDLLSIFEENLF 1362
short ----- 722

```

Altogether, we believe we have overexpression of the desired BRD4 isoforms in our cell line models. We have added the immunoblot result as Figure S4.

3. Rearranged BRD4 locus could also affect the expression of another BRD4 short isoform, termed short form b - BRD4-S(b). It would be nice if the authors could also analyze whether recurrent focal BRD4 deletions detected in various cancer types may also affect BRD4-S(b) transcript levels.

We have focused our analysis and experiments on the two most dominant isoforms in the PCAWG data, BRD4-short isoform BRD4-203, and the BRD4-long isoform BRD4-204. The reviewer is correct to point out that there is indeed another BRD4-short isoform (BRD4-202; ENST00000360016.5) that has been annotated in the PCAWG expression dataset. Although the absolute expression of this isoform is significantly less than BRD4-203, we do indeed see that focal deletions have the same effect on BRD4-202 as in the primary BRD4-203 short isoforms -- reducing copy-number adjusted expression. This is now described in the main text, and we have included analysis of this isoform into Figure 2C.

We further considered two other annotated protein-coding *BRD4* transcripts (ENST00000594841.1 and ENST00000601941.1) with transcript-level expression data in the PCAWG dataset. However, the median expression in breast, ovarian and endometrial cancers was negligible (median FPKM: ENST00000594841 = 0.09, ENST00000601941 = 0.18), and we did not include these isoforms in further analyses.

4. BRD4 overexpression has been detected in many cancer types and correlates with oncogenic potential. This appears to be contradictory to the view conveyed in the current study. Since dose response is critical in biological readout, the authors may need to carefully define what levels of "overexpression" refer to in their context.

We agree with the reviewer that *BRD4* overexpression can be an oncogenic driver, and the balance between oncogenicity and toxicity likely depends on the cellular context. For instance, we note that median *BRD4* expression varies across tumor types in the PCAWG dataset by 5-fold (median FPKM: 8.35 in ccRCC to 44.38 in ovarian cancer), suggesting that tolerance of *BRD4* is highly dependent on the tumor type.

To disentangle the effects of *BRD4* focal deletions, background copy-number (e.g. amplification in the context of *CCNE1* amplification) and tumor type, we have now implemented linear regression using the PCAWG data to model the effect sizes of each of these factors (**Reviewer Figure 2**). We find that tumor type is the most important factor influencing absolute *BRD4* expression levels (with ovarian cancer having the largest positive effect; regression coefficient: 26.1, 95% CI: 22.5-29.7), but that *BRD4* focal deletions further modulate absolute expression to nearly the same extent as a loss of two copies (regression

coefficient: -6.3, 95% CI: -13.8 - 1.3). This suggests that the extent to which absolute *BRD4* expression is beneficial or toxic is highly tissue dependent, but remains strongly modulated by the genomic status within its tissue context. The results of this model are now described in Figure 2E of the revised manuscript.

5. Immunoblotting performed using lysates prepared in RIPA buffer containing 150 nM NaCl may not be right.

We thank the reviewer for pointing out that the correct unit for NaCl's concentration is 150 mM instead of 150 nM. We have corrected this in the 'Methods' section of the revised manuscript.

Reviewer #2:

In the manuscript titled "Recurrent breakpoints in the BRD4 locus reduce toxicity associated with gene amplification", the authors using genomic tools and OVSAHO ovarian cancer cells experiments identified and elucidated a 'Goldilocks' model of BRD4 gene regulation, where too high or too low levels of expression have a negative effect on cellular fitness. In BRD4-amplified tumors, focal deletions in gene regulatory regions serve as a mechanism to decrease and restore gene expression to levels that are tolerated and beneficial for proliferation. This represents a novel mechanism of gene-dosage compensation in human cancers.

It's a very interesting study that provides the first experimental evidence for a novel mechanism by which genetic alterations drive cancer. However, some points should be addressed:

1. The amplification curves of the +GFP group in Figure 3F and Figure 4F are very similar, please check whether the same set of data was used. In addition, the amplification curves of the +BRD4-long and +BRD4-short groups in Figure 3F and Figure 5B are also very similar, please check whether the same set of data was used as well.

The amplification curves pointed out by the reviewer are indeed the same, as we originally evaluated the effect of BRD4 isoform overexpression and CRISPR-Cas9-mediated cutting within the same experiment. The data were initially separated to simplify comparisons between BRD4- and GFP-overexpressing ORFs, as well as sgBRD4 versus sgGFP control on cellular proliferation. We thank the reviewer for this observation and have now revised this figure to present all conditions simultaneously (**Reviewer Figure**

3), as well as to highlight relevant comparisons (Reviewer Figures 4a-d). The updated growth curve with all conditions is presented as Figure S5 of the revised manuscript, and Figures 3F, 4F and 5B have been updated accordingly.

2. In Figure 5A, the authors first simulated focal deletions in the BRD4 regulatory region by CRISPR-Cas9 technology in OVSAHO cells, and then transfected BRD4-long and BRD4-short, and found that these focal deletions could rescue ovarian carcinoma cells from the toxic effects associated with gene overexpression. I would be also very interested if the order of process was changed, with transfection of BRD4-long and BRD4-short first to simulate the overexpression of BRD4-long and BRD4-short, and then the focal deletion of the BRD4 regulatory region was simulated by CRISPR-Cas9 technology. This order seems to shed more light on whether focal deletions protect ovarian cancer cells from the toxic effects of gene overexpression.

We agree with the reviewer that it would be interesting to assess the order of events regarding *BRD4* overexpression and focal deletions. The richness of the PCAWG dataset, which includes allelic copy-number calls, provides a means for determining the most likely temporal ordering for a majority of cases. For instance, a tumor with an allele exhibiting both loss of heterozygosity and an amplification suggests that the deletion likely came first. Conversely, an allele with a deletion resulting in a non-zero copy number implies the amplification occurred first. If the amplification and deletion affected different alleles, we deem the sample "ambiguous" with respect to event order. To address this point, we analyzed allelic copy number data from PCAWG tumors to determine the most parsimonious order of events. Six cases were ambiguous due to insufficient allelic information in the case of very short deletions or in the

case of deletions affecting a different allele than the amplification. In the 13 samples with focal deletions in *BRD4* and sufficient allelic information, we identified an enrichment for amplifications occurring first (in 11 samples) compared with deletions occurring first (2 samples; $p = 0.01$; binomial test; **Reviewer Figure 5**). This suggests that *BRD4* focal deletions may confer an additional fitness advantage to *CCNE1* amplified tumors, rather than serve as a necessary antecedent to these amplifications. This analysis is now included as Figure S3 in the revised manuscript.

In our experimental design, we aimed to have overexpression of *BRD4* and CRISPR-Cas9-mediated focal deletions happen simultaneously. The reason for this is based on our observations that *BRD4* overexpression alone (**Figure 3F**), or *BRD4* focal deletions alone (**Figure 4F**), significantly reduce cell viability - making it technically infeasible to clearly time one event before the other. Indeed, we observed that only when over-expression and sgRNA cutting were timed to occur at approximately the same time (or within a few cell cycles), cells would continue to maintain viability (**Figure 5**). This is consistent with the "goldilocks" model for *BRD4* expression. To achieve this, we had to account for the different timescales for each of these perturbations: efficient sgRNA cutting takes 14 days whereas *BRD4*-long/-short overexpression is achieved in 2-3 days. We therefore first transduced cells with CRISPR-Cas9 sgRNAs, followed by *BRD4* overexpression, to limit the number of cell cycles between *BRD4* over-expression and focal deletions.

number for samples with both amplification and deletion at *BRD4* suggests there may be positive selection for *BRD4* deletions in *CCNE1* amplified tumors (binomial test, $p=0.01$). Red and blue distinguish the major and minor alleles, and black arrows denote SV breakpoints called by the ICGC-PCAWG consensus SV pipeline. In several instances, the deletion effect is too small to register a change in copy number but is confirmed to affect read depth (plotted in **Figure S2**). Visible copy losses are occasionally unaccompanied by a consensus-called breakpoint.

3. The paper uses a variety of statistical methods, just some of which are described, please add a general description of statistical methods for others.

We thank the reviewer for pointing out that a description of statistical analyses is missing in the manuscript. We have now updated the Methods section with a description of statistical analyses used throughout the paper.

4. Please note the individual writing errors.

(1) In line 327, "RPMI 1460 Medium" should be "RPMI 1640 Medium".

(2) In line 329, "1% Pen Strep" should be "1% Pen/Strep".

(3) In line 330, "Advanced DMEM (cat. 330 no. 12491015, ThermoFisher Scientific) medium supplemented with 10% FBS" should be changed to "Advanced DMEM medium (cat. 330 no. 12491015, ThermoFisher Scientific) supplemented with 10% FBS". In addition, please check if 1% Pen/Strep is also added here.

(4) In line 345, "ug/mL" should be "µg/mL".

We thank the reviewer for a thorough revision of the Methods section. We have now corrected each of these individual errors, including the addition of 1% Pen/Strep when describing the conditions under which HEK-293T cells were cultured.

Referees' report, second round of review

Reviewer 1

MS#: CELL-GENOMICS-D-24-00463R1

Authors: Wala, Dalin, Webster, Shapira, Busanovich, Sarmashghi, Beroukhim, Bandopadhyay, and Rendo

This revised submission addressed most of my earlier comments. However, the wording related to the two short isoforms remains ambiguous, which would be helpful to further clarify:

1. The dominant BRD4-203 short isoform, referred to by the authors, has been previously defined as BRD4 short isoform a, i.e., BRD4-S(a), in the literature (Ref. 17). Likewise, the secondary BRD4-202 short isoform has been named BRD4-

S(b). Without causing confusion, it is better to follow the published nomenclature when referring to each specific isoform.

2. Both BRD4-S(a) and -S(b) antibodies have been described (Ref. 17). Without validation, the identity of the "BRD4-short" band in Figure 4A is questionable and may represent a predominant degradation product from the long isoform as mentioned in my previous comment.

3. Differences in BRD4 RNA isoforms are not equivalent to the fold changes of the protein isoforms, as BRD4 transcripts are also subject to post-transcriptional regulation (e.g., see Fig. 3D vs. 3E). It is important to clearly specify in the text whether specific RNA or protein levels are monitored in the assays.

4. BRD4-short, referred to in Figures 3-5, should be clearly specified as BRD4-S(a) or defined as BRD4-S earlier in the text (e.g., Figure 3 onward). Without specification, the results may be interpreted as BRD4-S(b) as occurred frequently in the literature.

5. Focal deletions of intron 1 and exon 1, likely situated upstream of the coding sequences of BRD4-L and BRD4-S(a), probably won't affect the coding capacity of the transcribed transcripts. If so, the translated protein should maintain the functional properties of the wild-type protein. The authors may want to comment on this in the text.

6. The long and short isoforms mentioned in the original abstract are simply replaced by BRD4, which slightly reduces the significance and impact of the current work. The clarity of BRD4 isoforms is important for the study.

7. Numbers "15" in line 84 and "23" in line 136 do not add up with individual counting. The authors should check the numbers again, which perhaps ought to be 16 and 22, respectively.

Reviewer 2

Thank you for your diligent work in addressing the issues I raised . Based on the fact that all the provided issues have been resolved, this manuscript can be accepted for publication.

Authors' response to the second round of review

Reviewer #1:

Reviewer #1: MS#: CELL-GENOMICS-D-24-00463R1
Authors: Wala, Dalin, Webster, Shapira, Busanovich, Sarmashghi, Beroukhim, Bandopadhyay, and Rendo

This revised submission addressed most of my earlier comments. However, the wording related to the two short isoforms remains ambiguous, which would be helpful to further clarify:

1. The dominant BRD4-203 short isoform, referred to by the authors, has been previously defined as BRD4 short isoform a, i.e., BRD4-S(a), in the literature (Ref. 17). Likewise, the secondary BRD4-202 short isoform has been named BRD4-S(b). Without causing confusion, it is better to follow the published nomenclature when referring to each specific isoform. We agree with the reviewer that it is important to maintain consistency in the nomenclature used to refer to the different BRD4 isoforms. We have now changed the manuscript text and figures to refer to BRD4-long as BRD4-L and the two BRD4-short isoforms as follows: the dominant BRD4-S(a) isoform (Ensembl name BRD4-203), and the secondary BRD4-S(b) isoform (Ensembl name BRD4-202).

2. Both BRD4-S(a) and -S(b) antibodies have been described (Ref. 17). Without validation, the identity of the "BRD4-short" band in Figure 4A is questionable and may represent a predominant degradation product from the long isoform as mentioned in my previous comment.

We agree with the reviewer that in Figure 4A of the manuscript we have used a BRD4 antibody that is not isoform-specific, and we therefore cannot conclude that the lower band present in OVSAHO cells (prior to transduction with BRD4-L or BRD4-S(a) ORFs) corresponds to the expression of a BRD4-S isoform. We have now modified the nomenclature in this immunoblot figure to clarify this point (Reviewer Figure 1).

However, the immunoblot results from Figure 3A strongly suggest that we are achieving overexpression of our BRD4-S(a) ORF construct in transduced OVSAHO cells, a band which is not present in parental cells nor GFP control or cells transduced with the BRD4-L ORF construct. Of course, we cannot disregard that OVSAHO cells express BRD4-L degradation products with potential functional roles. Nevertheless, in the context of our isogenic overexpression models, these potential degradation products are unlikely to drive the differences in effect size that we observe in our downstream analyses (i.e. growth curves), as the levels of endogenous BRD4-L (and potential degradation products) should be controlled for across the different models. Similarly to what the authors describe in Ref 17 (Wu et al. *Molecular Cell* 2020), our cells express low levels of BRD4-S(b), which impairs the validation of this antibody in our current experimental setup. We have now addressed this comment in two ways: i) we have modified the immunoblot in Figure 3A to clarify that we are detecting BRD4 protein expression in an isoform unspecific manner (Reviewer Figure 2), and ii) we have added a new paragraph in the discussion describing the possibility that our OVSAHO-derived cell line models express BRD4-L degradation products with functional cellular roles, and our limitation in detecting these with our current reagents.

3. Differences in BRD4 RNA isoforms are not equivalent to the fold changes of the protein isoforms, as BRD4 transcripts are also subject to post-transcriptional regulation (e.g., see Fig. 3D vs. 3E). It is important to clearly specify in the text whether specific RNA or protein levels are monitored in the assays.

We thank the reviewer for this comment. We now clarify throughout our different experiments whether we quantify and compare mRNA or protein isoform levels across OVSAHO-derived cell line conditions.

4. BRD4-short, referred to in Figures 3-5, should be clearly specified as BRD4-S(a) or defined as BRD4-S earlier in the text (e.g., Figure 3 onward). Without specification, the results may be interpreted as BRD4-S(b) as occurred frequently in the literature.

We have replaced "BRD4-short" with BRD4-S(a) in all our figures (and manuscript text) to specify that this is the short isoform being functionally studied across experiments.

5. Focal deletions of intron 1 and exon 1, likely situated upstream of the coding sequences of BRD4-L and BRD4-S(a), probably won't affect the coding capacity of the transcribed

transcripts. If so, the translated protein should maintain the functional properties of the wild-type protein. The authors may want to comment on this in the text.

In our OVSAHO cell line models, we observe that CRISPR-Cas9-mediated cutting of region 1 and regions 2 on *BRD4*'s intron 1 (corresponding to the genomic location of the focal deletions observed in human PCAWG data) do affect the levels of *BRD4* mRNA expression (Figure 4E). We believe this is because such focal deletions span gene regulatory regions. On the other hand, the reviewer is right in pointing out that we did not assess the impact of these deletions on the translated protein in our cell line models. We have now added a sentence to comment on this in our results section.

6. The long and short isoforms mentioned in the original abstract are simply replaced by *BRD4*, which slightly reduces the significance and impact of the current work. The clarity of *BRD4* isoforms is important for the study.

We agree with this comment and have now modified the abstract to reflect how we have generated reagents and cell line models to evaluate the impact of different *BRD4* isoforms on cellular fitness.

7. Numbers "15" in line 84 and "23" in line 136 do not add up with individual counting. The authors should check the numbers again, which perhaps ought to be 16 and 22, respectively.

We thank the reviewer for noticing these mistakes. We have now updated the manuscript text to reflect that we find recurrent *BRD4* focal deletions in 16 tumors, and that, in a follow-up analysis, 23 genes show a significant differential expression between *BRD4* focal deletion and non-deleted cohorts.

Reviewer #2:

Thank you for your diligent work in addressing the issues I raised . Based on the fact that all the provided issues have been resolved, this manuscript can be accepted for publication.

We thank the reviewer for their comments, which improved the content of the manuscript.